# In vivo engineered extracellular matrix scaffolds with instructive niches for oriented tissue regeneration

Meifeng Zhu[1,2,3], Wen Li[1], Xianhao Dong[1], Xingyu Yuan[1], Adam C. Midgley [1], Hong Chang[1], Yuhao Wang[2], Haoyu Wang[2], Kai Wang[1]*, Peter X. Ma [4], Hongjun Wang[2]* & Deling Kong[1,3]*

Implanted scaffolds with inductive niches can facilitate the recruitment and differentiation of host cells, thereby enhancing endogenous tissue regeneration. Extracellular matrix (ECM) scaffolds derived from cultured cells or natural tissues exhibit superior biocompatibility and trigger favourable immune responses. However, the lack of hierarchical porous structure fails to provide cells with guidance cues for directional migration and spatial organization, and consequently limit the morpho-functional integration for oriented tissues. Here, we engineer ECM scaffolds with parallel microchannels (ECM-C) by subcutaneous implantation of sacrificial templates, followed by template removal and decellularization. The advantages of such ECM-C scaffolds are evidenced by close regulation of in vitro cell activities, and enhanced cell infiltration and vascularization upon in vivo implantation. We demonstrate the versatility and flexibility of these scaffolds by regenerating vascularized and innervated neo-muscle, vascularized neo-nerve and pulsatile neo-artery with functional integration. This strategy has potential to yield inducible biomaterials with applications across tissue engineering and regenerative medicine.

[1] College of Life Science, Key Laboratory of Bioactive Materials (Ministry of Education), State Key Laboratory of Medicinal Chemical Biology, Xu Rongxiang Regeneration Life Science Center, Nankai University, 300071 Tianjin, China. [2] Department of Biomedical Engineering, Stevens Institute of Technology, Hoboken, NJ 07030, USA. [3] Tianjin Key Laboratory of Medical Epigenetics, Tianjin Medical University, Tianjin, China. [4] Department of Biologic and Materials Sciences, Department of Biomedical Engineering, Macromolecular Science and Engineering Centre, Department of Materials Science and Engineering, University of Michigan, Ann Arbor, MI 48109, USA. *email: wkcs424@163.com; hongjun.wang@stevens.edu; kongdeling@nankai.edu.cn

Trauma, disease and congenital abnormalities often lead to tissue dysfunction or organ loss, requiring prompt restoration of the lost functions. In view of the high risk of donor site morbidity from autografting and the limited donor tissues and organs for allotransplantation[1], emerging attention has shifted to engineer biomaterials that can trigger the body's innate regenerative mechanism to restore, maintain, or improve the damaged tissue[2,3]. Accumulative evidence showed that well-designed scaffolds indeed induce the formation of functional neo-tissue, in vivo, by harnessing endogenous regenerative capacity[4–6].

To achieve the unique physiological functions, many tissues in our body are anisotropic with distinct spatial organisation of their extracellular matrix (ECM) and residing cells. For example, in oriented tissues (e.g., muscle, nerve, artery), obvious alignment of cells and ECM are observed[7,8]. Such an organisation modulates the mechanical properties of tissues, and impacts upon cellular functions, such as cytoskeleton reorganisation, integrin activation, gene expression and ECM remodelling[7,8]. Thus, scaffold-guided cell arrangement is a vital factor towards functional regeneration and maturation of oriented tissues[8,9]. Furthermore, in vivo recruitment of tissue-forming cells into scaffolds is closely regulated by their physicochemical properties, e.g., pore size, porosity, bioactivities, stiffness, etc[10,11]. Significant progresses have been made to fabricate scaffolds for tissue regeneration using synthetic or natural materials[12,13]. In contrast to large lot variation, low mechanical strength and uncontrollable degradation of natural materials (e.g., collagen, chondroitin and hyaluronic acid)[14–16], synthetic polymers offer better control of the physicochemical properties and processability. However, oftentimes the synthetic polymers may elicit unwanted inflammation and lead to undesirable tissue formation as a result of acidic degradation by-products[14,17]. In recognition, increasing efforts also shift to develop ECM scaffolds by decellularizing the cultured cells or native tissues[16]. While demonstrating their biological superiority in terms of preferred cellular activities and minimised immune responses[18], the ECM scaffolds derived from native tissue usually exhibited noted advantages over those from cultured cells, especially in consideration of the architectural/compositional complexity and mechanical stability[13,19,20]. Upon in vivo implantation, the intrinsic tubular networks (i.e., vessel and nerve) preserved in tissue-derived ECM scaffolds are beneficial for tissue ingrowth and vascularisation[21,22]. However, it is always a technical challenge to maximally retain these channel structures[13,16]. Furthermore, some native tissues like tendon and ligament inherently having dense ECM would inhibit cell infiltration and lead to deficient tissue regeneration[23]. Therefore, development of ECM scaffolds with three-dimensional hierarchical pore structure would be essential for endogenous cell infiltration and functional tissue regeneration in situ[6,8,24].

Certain in vivo environments such as peritoneal cavity or subcutaneous pocket can be used as bioreactors to generate fibrous capsules around the implanted templates (e.g., silicone rod or sheet) and then the fibrous capsules are used as grafts to support tissue repair and islet cell transplantation[25–31]. However, these grafts are more prone for autologous transplantation, and their inherent cellular components and dense ECM structure normally limit the remodelling by local microenvironment. Previous studies demonstrated the utility of polymer-based matrices as pore-forming templates to generate porous ECM scaffolds in vitro[20,32,25]. Inspired by these studies, we proposed a new strategy to generate ECM scaffolds with aligned microchannels (Fig. 1), following the steps of: (1) design and fabrication of sacrificial templates (membranous or tubular) consisting of aligned polymeric microfibers; (2) implantation of the templates into rat subcutaneous pockets for cellularization and tissue formation; (3) removal of the polymeric template and cellular component.

Following the aforementioned steps, ECM scaffolds with aligned microchannels (ECM-C) were generated. The supportiveness of ECM-C for in vitro cell migration and spatial organisation and in vivo guided tissue formation with favourable vascularisation and immunomodulation was respectively investigated. To demonstrate the generality and applicability of our proposed approach, ECM-C with two configurations, i.e., sponge or tube, were specifically fabricated and respectively used to regenerate three representative oriented tissues including rat tibialis anterior muscle with volumetric muscle loss, sciatic nerve and abdominal artery of a critical size defect. Comprehensive analyses were conducted to evaluate the efficacy of ECM-C in regenerating oriented tissues.

## Results

**Preparation and characterisation of ECM-C scaffolds**. Following the established fabrication method[33], aligned PCL microfibers with diameters of 141.8 ± 5.2 μm (mean ± s.e.m.) were collected and used for preparing the membrane templates (thickness = 1.5 mm). Upon subcutaneous implantation into rats for 4 weeks, the PCL microfiber membranes were harvested and newly formed fibrous tissue occupied all the inter-fibre space (Fig. 2a). Cells within the interstitial space of aligned microfibers (Supplementary Fig. 1a) synthesised abundant collagen-dominated ECM as confirmed by Sirius red staining (Supplementary Fig. 1b). A majority of these cells were positive for α-smooth muscle actin (α-SMA) (Supplementary Fig. 1c), whereas few were stained for the inflammatory macrophage cell marker, CD68 (Supplementary Fig. 1d). Upon removal of PCL template and cellular component, translucent ECM-C was obtained with reservation of their original shape and dimensions. Scanning electron microscopy (SEM) examination of ECM-C showed the presence of uniformly distributed parallel microchannels with an average diameter of 146.6 ± 6.9 μm based on the longitudinal cross-section (Fig. 2a). Fibrosis tissue capsules generated on the surface of dense silicone membranes coated with a thin layer of PCL after one-month subcutaneous implantation was used as controls. However, such tissue capsules could not maintain their original shape either before or after decellularization (Supplementary Fig. 2a, b and Fig. 2a). Meanwhile, the decellularized tissue capsules had a relatively smooth surface with no obvious pores (Fig. 2a). MicroCT scanning of different positions further affirmed uniform and interconnected oriented microchannel structures throughout ECM-C, and no evident pore structure across the control scaffolds (MicroCT, Fig. 2a, Supplementary Fig. 3a, b, Supplementary Movie 1 and 2). Based on the MicroCT results, the porosity of ECM-C was significantly higher than control scaffolds (74.4 ± 2.1% vs. 26.7 ± 5.4%). Also, ECM-C exhibited noted anisotropy in comparison to control scaffolds (0.89 ± 0.12 vs. 0.14 ± 0.09). Gel permeation chromatography measurement confirmed complete removal of PCL microfibers following the polymer leaching process (Fig. 2b). While no residual nuclei were detected after decellularization as validated by DAPI staining, H&E staining showed that both types of scaffolds retained substantial ECM components (Fig. 2c, d). Moreover, the DNA content in both scaffolds was substantially decreased when SDS and DNase/RNase were used in combination with the decellularization protocol (Fig. 2e). Residual DNA content was 32.3 ± 9.1 and 33.4 ± 12.4 ng per mg for ECM-C and control scaffolds, respectively, much lower than the minimal criteria for acellular products (50 ng per mg)[34]. Both scaffolds were mainly composed of collagen, elastin and sulphated glycosaminoglycans (sGAG), with the collagen fibres arranged in an aligned fashion in ECM-C and distributed randomly in control scaffolds, as shown in the transmission electron microscopy (TEM) and histological images (Fig. 2f). A marked amount of ECM components was still retained after decellularization, i.e., the

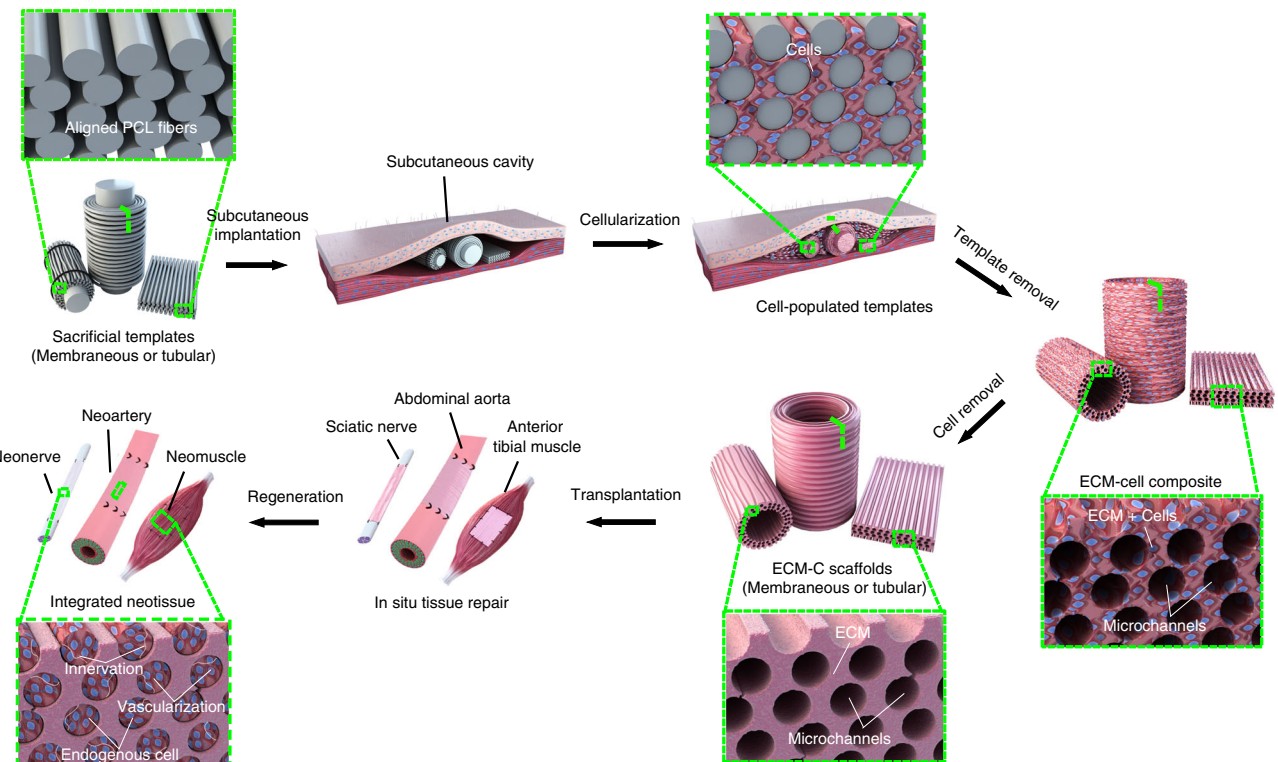

**Fig. 1** Sketch map of generation and application of ECM scaffold. Schematic illustration of the key steps leading to the formation of ECM scaffolds with aligned microchannels for demonstrated utility in guided regeneration of muscle, nerve and artery

content of collagen in ECM-C was higher than that of control group, and elastin and sGAG content were comparable between both groups (Fig. 2g). The maximum stress of ECM-C was noticeably higher than that of control scaffolds, while no significant differences were observed in elastic modulus or elongation at break between both scaffolds (Supplementary Fig. 4a–c). Furthermore, the suture retention of ECM-C was higher than that of control scaffolds ($2.4 \pm 0.3$ N vs. $2.0 \pm 0.1$ N), and both scaffolds had the required strength for microsurgical anastomosis using 9-0 sutures ($0.69 \pm 0.04$ N).

**Cell behaviours regulated by ECM-C scaffolds in vitro**. The guiding effect of ECM-C on cells was investigated in vitro in comparison to that of controls. Three types of cells (rat skeletal muscle cell line L6, Schwann cell line RSC96 and vascular smooth muscle cell line A10) were respectively seeded onto ECM-C and control scaffolds. After 1-day culture, cells grew longitudinally along the microchannels throughout ECM-C, whilst the cells distributed randomly across the control scaffolds (Fig. 3a–c). The cell density increased in both scaffolds at 3 days, and the orientation of cell growth on ECM-C became even more obvious than one-day culture (Fig. 3a–c). Based on DAPI staining, the cell nuclear shape on control scaffolds was typically round while on ECM-C they exhibited the elongated shape at 3 days (Fig. 3d). As a result, the circularity of L6, RSC96 and A10 cells on ECM-C was significantly lower than that on control scaffolds at 3 days, indicating that ECM-C possessed a strong guiding effect (Fig. 3e). The migration behaviours of L6, RSC96 and A10 on both scaffolds were also examined. The trajectory of L6, RSC96 and A10 cells on ECM-C exhibited a radial pattern; in contrast, these cells remained close to the initial position on control scaffolds (Fig. 3f). Moreover, Euclidean distance of L6, RSC96 and A10 cells on ECM-C was significantly longer, and the velocity was obviously higher than that on control scaffolds (Fig. 3g–i). CCK-8 assay

demonstrated that all three types of cells cultured on ECM-C proliferated better than that on control scaffolds, and significant difference was observed at 3 and 7 days (Supplementary Fig. 5a, b, c). At 7 days, live/dead cell assays showed higher cell viability on ECM-C than on control scaffolds, suggesting that ECM-C might possess a high level of comfortable living space for residing cells (Fig. 3j, k). Real-time PCR analysis showed that gene expression of *Myh* and *Myog* in L6 on ECM-C was higher than that on control scaffolds, despite no obvious difference (Fig. 3l). Gene expression of *Nrcam*, a marker gene only expressed in immature Schwann cells (SCs), showed significant decrease on ECM-C than that on control scaffolds (Fig. 3l). Gene expression of *Krox20*, an essential transcription factor for SC myelination, in RSC96 cells was significantly up-regulated on ECM-C compared to control group, implying that the existence of microchannels in ECM-C favoured SC myelination (Fig. 3l). Gene expression of *Ngf*, one of the most important neurotrophins in nervous systems, in RSC96 cells on ECM-C was also significantly elevated compared to that on the control scaffolds, suggesting that the guidance cues of parallel microchannels in ECM-C possess a stronger ability to enhance neurotrophin secretion (Fig. 3l). Expression of *Acta 2* and *Myh 11* genes in A10 cells on ECM-C seemed higher than that on the control scaffolds despite no statistical difference (Fig. 3l). Taken together, all these data confirmed that ECM-C were able to induce cell elongation, facilitate cell migration, maintain cell viability, promote cell proliferation, enhance functional gene expression, and provide a more favourable microenvironment for oriented tissue regeneration than control scaffolds.

**Performance of ECM-C scaffolds in vivo**. Both ECM-C and control scaffolds were subcutaneously implanted into rats to investigate their capability for cellularization, vascularisation and immunoregulatory effect. Both DAPI and H&E staining

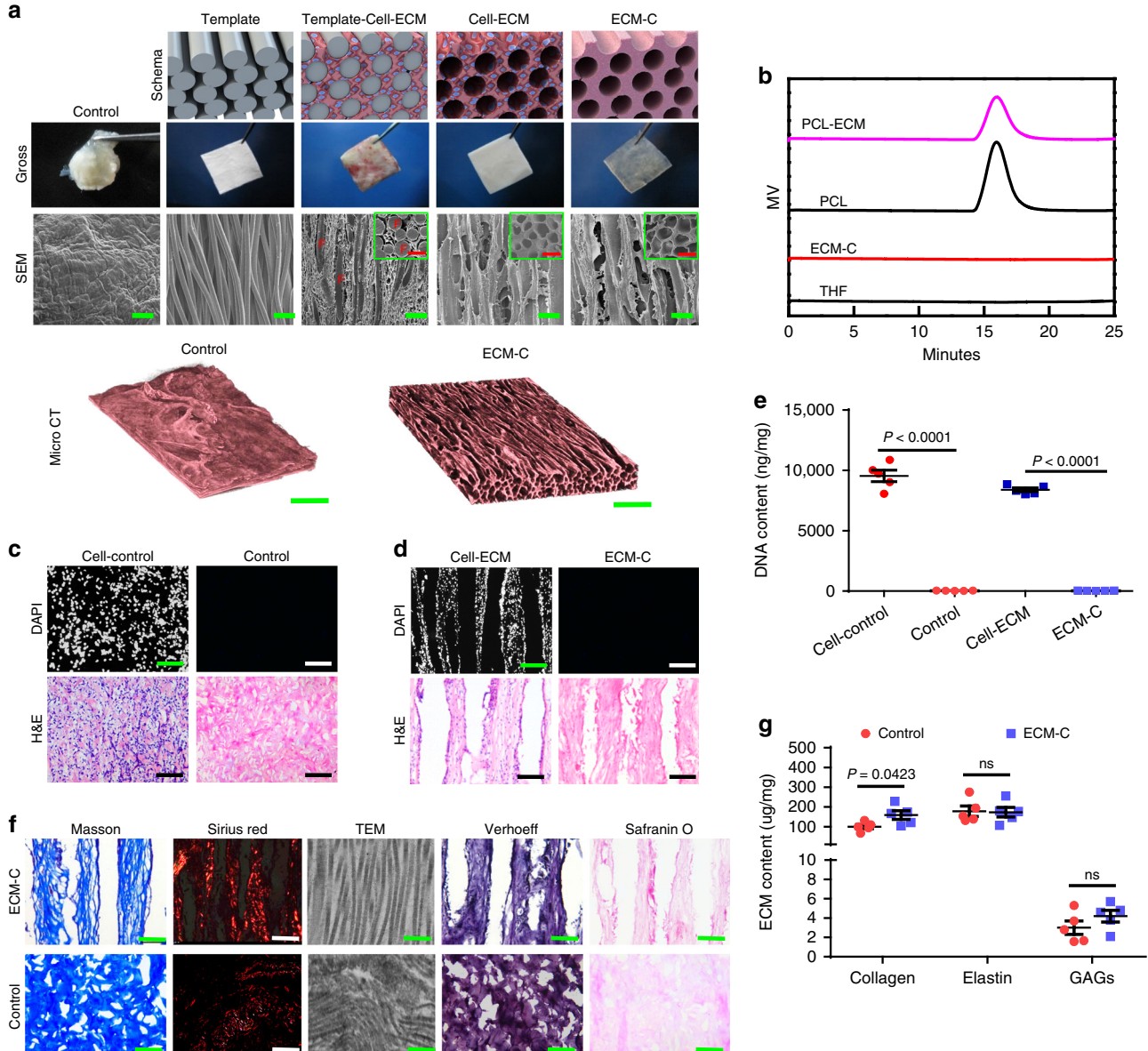

**Fig. 2** Fabrication and characterisation of ECM-C and control scaffolds. **a** The schematic diagram, gross morphology (optical imaging) and the microstructure (SEM) of PCL fibre template, template-cell-ECM, cell-ECM, ECM-C and control group during the preparation process; micro-computed tomography (microCT) showing three-dimensional macro and micro structure of ECM-C and control scaffolds. Inset: SEM examination of the transverse sections. F indicates the PCL microfiber. **b** Gel permeation chromatography analysis showing complete removal of PCL. **c, d** DAPI and H&E staining of undecellularized control, control scaffolds, cell-ECM composite, and ECM-C scaffolds. **e** DNA contents of cell-ECM composite, ECM-C, undecellularized control and control group, the samples were treated with 1% SDS containing DNase and RNase (n = 5). **f** Masson trichrome staining, Sirius red staining, TEM, Verhoeff and safranin O staining showed the ECM components including collagen, elastin and glycosaminoglycan in ECM-C and control scaffolds. **g** Comparison of ECM contents between ECM-C and control group including collagen, elastin and glycosaminoglycans (n = 5). Bar heights and error bars represent means ± s.e.m. (t-test). Statistical analysis (ns = no significance). Scale bars: **a**, SEM images: 300 μm; Micro CT image: 500 μm; **c, d**, 100 μm; **f**, histology panels: 100 μm; TEM image: 200 nm

demonstrated that host cells could effectively infiltrated throughout ECM-C, whereas in the control scaffolds, most cells distributed around the marginal territory of the scaffolds (Fig. 4a, c). As a result, the cell population inside ECM-C was significantly higher than that of control scaffolds (Fig. 4b). Zoomed images of the H&E-stained sections also showed a high capillary density within ECM-C, as evidenced further by immunostaining for von Willebrand Factor (vWF). In contrast, much fewer capillaries were found in the interior of control scaffolds, and only some distributed around the scaffolds (Fig. 4d, e). Clearly, the number of capillaries inside ECM-C was significantly higher than that

within the control scaffolds (Fig. 4f). All the in vivo results demonstrated the superior cellularization and vascularisation capacity of ECM-C to control scaffolds. Macrophages (CD68-positive) can polarise between iNOS-positive pro-inflammatory macrophages (M1) and CD206-positive anti-inflammatory macrophages (M2). Immunostaining for these markers confirmed that CD206/CD68 and iNOS/CD68 double positive cells distributed throughout the internal and external surfaces of ECM-C. In contrast, the macrophages were mainly distributed around the peripheral boundary of control scaffolds (Fig. 4g). Image analysis showed that the number of CD68 and CD206-positive cells was

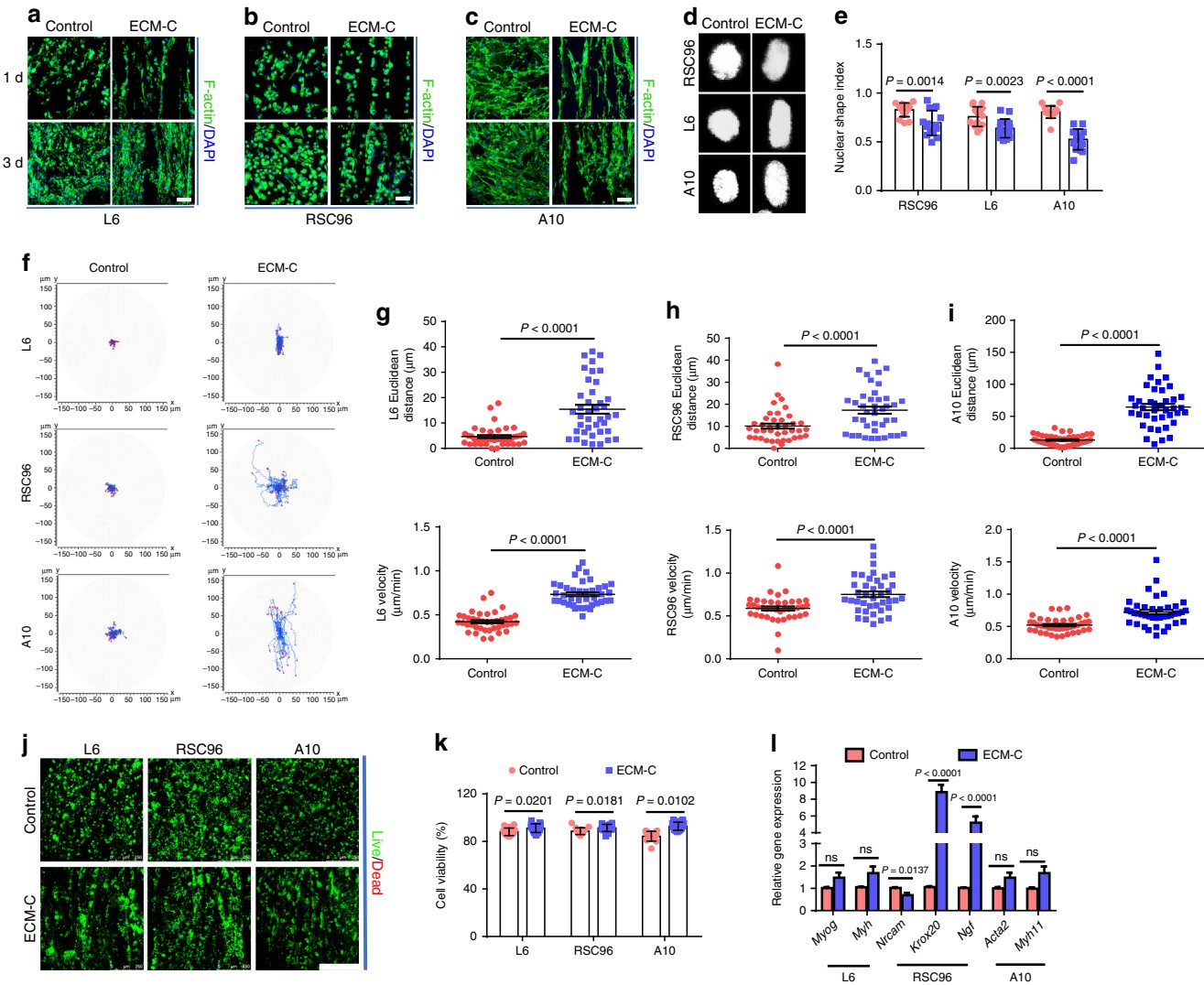

**Fig. 3** Cellular guiding effects of ECM-C and control scaffolds. **a–c** Skeletal actin fibres and nuclei of L6, RSC96 and A10 cells were respectively stained with fluorescein isothiocyanate (FITC) conjugated phalloidin and DAPI at 1 and 3 days. **d** Typical images of L6, RSC96 and A10 cells nuclei stained by DAPI. **e** The nuclear shape index of L6, RSC96 and A10 cells on different scaffolds (n = 15). **f** Migration traces of L6, RSC96 and A10 cells on different scaffolds. At least 40 cells were tracked for each group. **g–i** Euclidean distance and velocity of L6, RSC96 and A10 cells on different scaffolds (n = 45). **j, k** Live/dead staining and cell viability of L6, RSC96 and A10 cells on different scaffolds at 7 days. Live cells stained green and dead cells stained red (n = 15). **l** Real-time PCR analysis of *Myog* (myogenin) and *Myh* (myosin heavy chain), gene expression of L6 cells, *Nrcam* (neuronal cellular adhesion molecules), *Krox20* (early growth response 2), *Ngf* (nerve growth factor) gene expressions of RSC96 cells, *Acta2* (α-smooth muscle actin) and *Myh11* (smooth muscle myosin heavy chain) gene expression of A10 cells on different scaffolds after culture for 7 days (n = 5). Images and data are representative of n = 3 individual experiments, and bar heights and error bars represent means ± s.e.m. (*t*-test). Statistical analysis (ns = no significance). Scale bars: **a–c**, 50 μm, **j**, 250 μm

significantly higher in ECM-C than that in control scaffolds, whilst the number of iNOS-positive cells in ECM-C was close to that in control scaffolds (Fig. 4h). These data suggested that compared to the control scaffolds ECM-C had better immuno-modulatory and biocompatible properties.

**Skeletal muscle regeneration.** To demonstrate the capability of ECM-C in promoting the regeneration of oriented soft tissue with volumetric muscle loss, such as skeletal muscle defects, membranous muscle ECM-C with longitudinally aligned micro-channels (~150 μm in diameter) were specifically designed and fabricated. Size-matched membranous ECM-C and control scaffolds were fit and sewn to the tibialis anterior muscle defect (Fig. 5a). As noted, upon implantation the colour of control scaffolds remained white throughout the entire operation (Fig. 5b), whereas the muscle ECM-C turned immediately to red

by the blood (Fig. 5c). For control group, the defects were still visible even after one-month implantation and partially filled with white fascia-like tissue (Fig. 5b). For the ECM-C-treated group, the defects were almost completely filled with neotissue and the grafted area became indistinguishable from the surrounding native muscle except the boundary location marked by black sutures (Fig. 5c). H&E staining of the cross-sections of healed areas showed that a large number of cells infiltrated the interior of the scaffolds along with the deposition of large amounts of new ECM (Fig. 5d). This was not seen with control scaffolds, in which cell infiltration and new ECM formation were very limited (Fig. 5d). This was further confirmed by Masson trichrome staining, showing the remainder of high-density collagen located on the defect surface (Fig. 5d). Conversely, only a small fraction of collagen fibres scattered throughout the defect treated with ECM-C (Fig. 5d). As quantified, the collagenous area of control

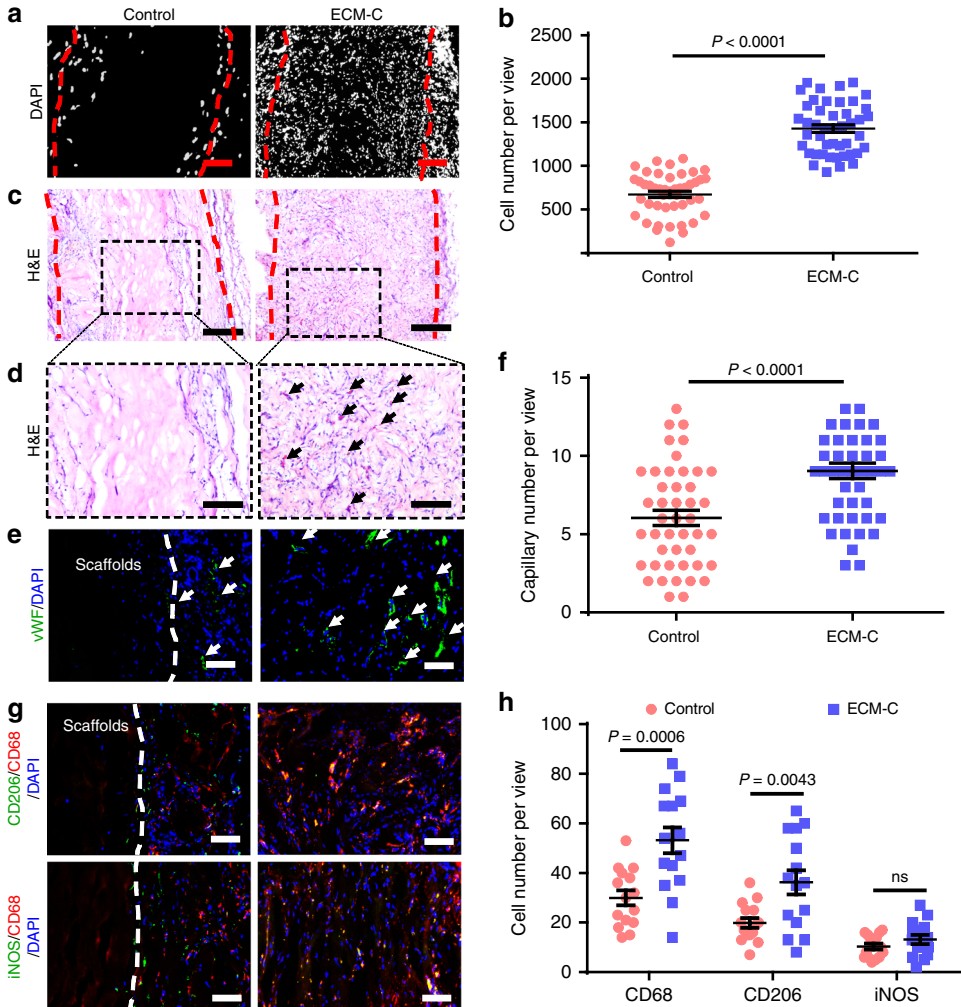

**Fig. 4** Cell infiltration, vascularisation and immunogenic properties of ECM-C and control scaffolds in a rat subcutaneous implantation model at 2 weeks. **a** DAPI staining of the cross-sections showing cell infiltration within the scaffolds. **b** Quantification of cells within the scaffolds ($n = 45$). **c–e** Optical images of cross-sections stained with H&E and immunofluorescently stained for vWF (green) showing cellularization and vascularisation within the scaffolds. The black and white arrows indicate the capillaries. The red dotted line indicates the edge of the scaffolds. **f** Quantification of capillaries within the scaffolds ($n = 45$). **g** Macrophages were detected by co-immunofluorescence staining for CD206 (green, anti-inflammatory)/CD68 (red, pan-macrophage), and iNOS (green, pro-inflammatory)/CD68 (red, pan-macrophage). White broken lines indicate the border of the scaffolds. **h** Quantification of CD68, CD206 and iNOS-positive cells within the scaffolds ($n = 15$). Images and data are representative of $n = 3$ individual experiments, and bar heights and error bars represent means ± s.e.m. (t-test). ($n = 5$ animals per group). Statistical analysis (no significance). Scale bars: **a**, 100 μm; **c**, 200 μm; **d**, 100 μm; **e**, **g**, 50 μm

group was significantly larger than that of ECM-C-treated group (Fig. 5e). Zoomed examination of the sections stained either with Masson trichrome or for desmin revealed the presence of a high density of neo-muscle fibres within ECM-C, while only sparse neo-muscle fibres were seen around the control scaffolds-treated defects (Fig. 5f). The average percentage of neo-muscle fibre area as well as the number of neo-muscle fibres in the control group was significantly lower than that in the ECM-C group (Fig. 5g, Supplementary Fig. 6a). More importantly, compared to the control group much more blood vessels were formed within the ECM-C groups as verified by both H&E staining and immuno-fluorescence staining for vWF (Fig. 5h, i). The presence of acet-ylcholine receptor (AChR) (α-BTX+) clusters on the neo-muscle fibres (desmin+), as well as nerve fibre (NF-09) contacts with α-BTX+ structures (i.e., neuromuscular junctions) within neo-muscle, indicated that ECM-C promoted nerve migration and integration within neo-muscle (Fig. 5j). However, this was not the case with control scaffolds (Fig. 5j, Supplementary Fig. 6b). These findings were consistent with the testing results of compound

muscle action potentials, of which the amplitude ratio of the ECM-C group was markedly higher than that of the control group (Fig. 5k).

**Nerve regeneration**. To demonstrate the capability of ECM-C in guided regeneration of peripheral nerve defects of critical size, tubular scaffolds with longitudinally oriented microchannels on the luminal surfaces and within the walls were designed and fabricated to achieve in situ nerve regeneration (Fig. 6a). The scaffolds obtained from solid silicone rods were used as controls. The obtained nerve ECM-C had a thick wall, enabling the maintenance of their structure and original shape; however, the relatively thin wall of control scaffolds easily collapsed to lose their shape upon hydration (Fig. 6b). MicroCT investigation confirmed uniform distribution of highly interconnected aligned micro-channels within the wall and on the luminal surface of ECM-C, and only a thin monolayer tube without obvious pores of the control scaffolds (Fig. 6b, Supplementary Movie 3 and 4). SEM

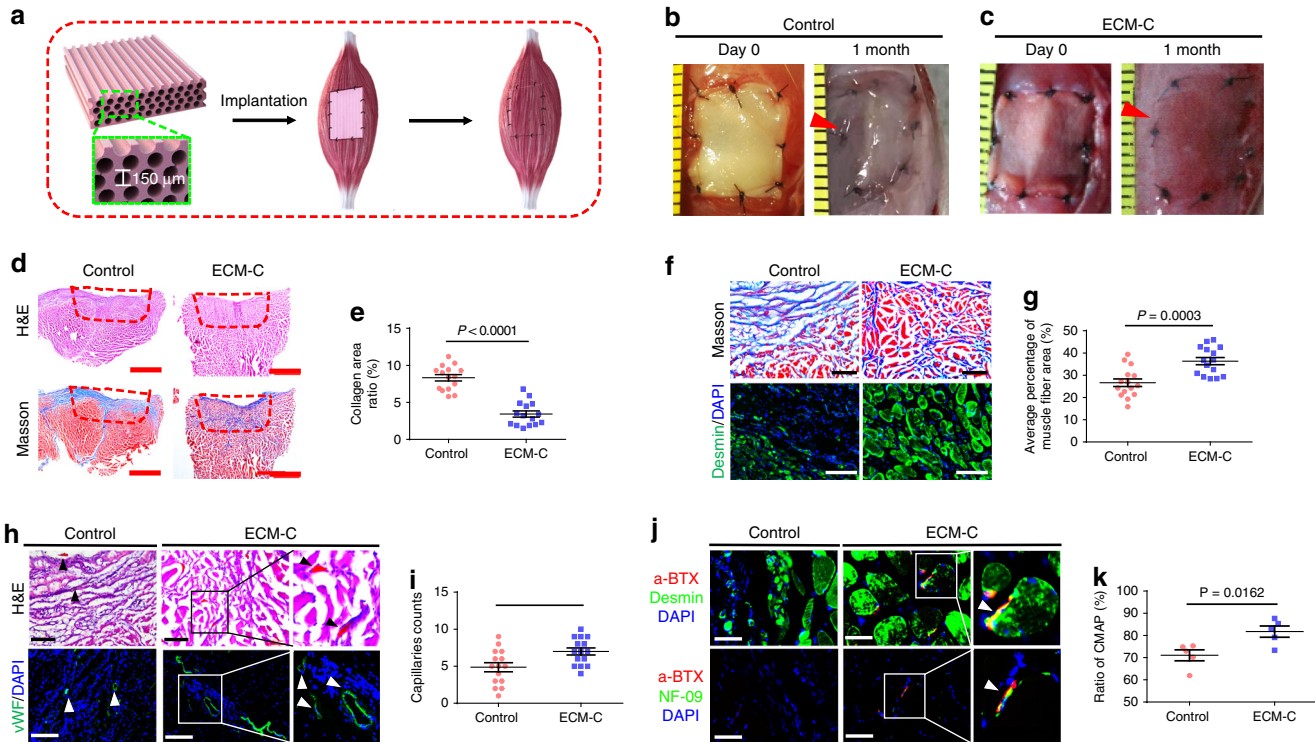

**Fig. 5** Muscle regeneration of the rat tibialis anterior (TA) muscle defects treated with ECM-C or control scaffolds. **a** Schematic illustration of muscle regeneration assisted by membranous ECM-C scaffolds with the microchannel diameter of ~150 μm. **b**, **c** Photographs of implantation and regeneration of muscle defects treated with control and ECM-C scaffolds at day 0 and 1 month, respectively. **d** Macroscopic view of regenerated TA muscle defects by staining the cross-sections with Masson trichrome and H&E. **e** Quantification of collagenous area in the cross-section of regenerated TA muscle defects (n = 15). **f** Microscopic images of the cross-sections of regenerated TA muscle at 1 month, stained with Masson trichrome and immunofluorescently stained for desmin. **g** Quantification of neo-muscle fibre per view (n = 15). **h** H&E staining and immunofluorescence staining of the cross-sections for vWF showing the formation of functional capillaries. Black and white triangles denote capillaries. **i** Number of capillaries per view (n = 15). **j** Demonstration of reinnervation by double immunofluorescence staining of neo-muscle fibres (desmin, green) and nicotinic acetylcholine receptors (α-bungarotoxin-tetramethylrhodamine) (α-BTX, red) or neurofilament (NF09, green) and α-BTX+ structure. White triangles indicate reinnervation within neo-muscle. **k** The amplitude ratio of regenerated muscle to uninjured TA muscle for the compound muscle action potentials (CMAPs) in both groups (n = 5). Bar heights and error bars represent means ± s.e.m. (t-test). (n = 5 animals per group). Scale bars: **b**, **c**, 1 mm; **d**, 2 mm; **f**, **h**, 100 μm; **j**, 50 μm

examination further revealed that parallel microchannels [diameter: 38 ± 6 μm (luminal surface) vs. 47 ± 8 μm (wall), p > 0.05] ran longitudinally through the wall and the luminal surface of nerve ECM-C while no visible pores were identified in the control scaffolds (Fig. 6b). The nerve ECM-C and control scaffolds were respectively implanted into rat sciatic nerve defects (15 mm) (Fig. 6c, d). After 3 months, the tubular ECM-C was able to remodel into neo-nerves with gross morphology and colour closely resembling the native nerves, but seemingly plumper than control scaffolds (Fig. 6c, d). H&E staining of the transverse sections of regenerated nerves showed that a high density of myelinated fibres existed not only in the lumen but also in the microchannels of ECM-C (Fig. 6e). ECM-C scaffold material was still present within the regenerated nerve defect (Fig. 6e). In contrast to the neo-nerve of ECM-C and autografts, only sparse immature nerve fibres distributed within the lumen of control scaffolds (Fig. 6e). Neo-capillaries were found in the interior and the wall of ECM-C and the number was surprisingly higher than that of autografts, especially in the wall microchannels (Fig. 6e–g). Very few capillaries could be seen inside or outside the control scaffolds (Fig. 6e–g). Furthermore, the regenerated nerve fibres were positive for neurofilament protein NF09 and calcium binding protein S100β (Fig. 6h). The density of NF09-positive fibres within ECM-C neo-nerve was significantly higher than that of control scaffolds, despite still lower than that of autograft (Fig. 6i). The CAMP ratios of ECM-C-regenerated neo-nerves were less than

that of autografts but significantly higher than control scaffolds (Fig. 6j, Supplementary Fig. 7). TEM examination of the mid-part of explanted scaffolds showed the presence of regenerated myelinated fibres with a compact and uniform structure in both ECM-C neo-nerves and autografts and these fibres were composed of clear, dense myelin sheathes and intact basal membranes. In control scaffolds, only thin myelinated fibres with small diameters were observed (Fig. 6k). The average axon diameter between ECM-C-regenerated neo-nerve and autografts exhibited no difference. The thickness of regenerated myelin sheathes in autografts was higher than that of ECM-C neo-nerves, and both were significantly higher than that of the control scaffolds (Fig. 6l, m). Sciatic nerve injury typically results in the atrophy of gastrocnemius muscle, one of the target tissues of sciatic nerves. The atrophy is usually measured by a decrease in muscle fibre size and muscle weight. Masson trichrome staining of the transverse section of gastrocnemius muscle revealed a large quantity of collagen deposited around the small-sized muscle fibres with the control groups. However, almost no tissue fibrosis could be detected in either nerve ECM-C or autografts group (Fig. 6n). Statistically, the muscle fibre size of ECM-C neo-nerve groups was significantly larger than that of control group despite smaller than that of the autograft group (Fig. 6o). Consistently, the wet weight of the gastrocnemius muscle in rats treated with ECM-C was lighter than that of autografts, however both groups were significantly heavier than that of control scaffold group (Fig. 6p).

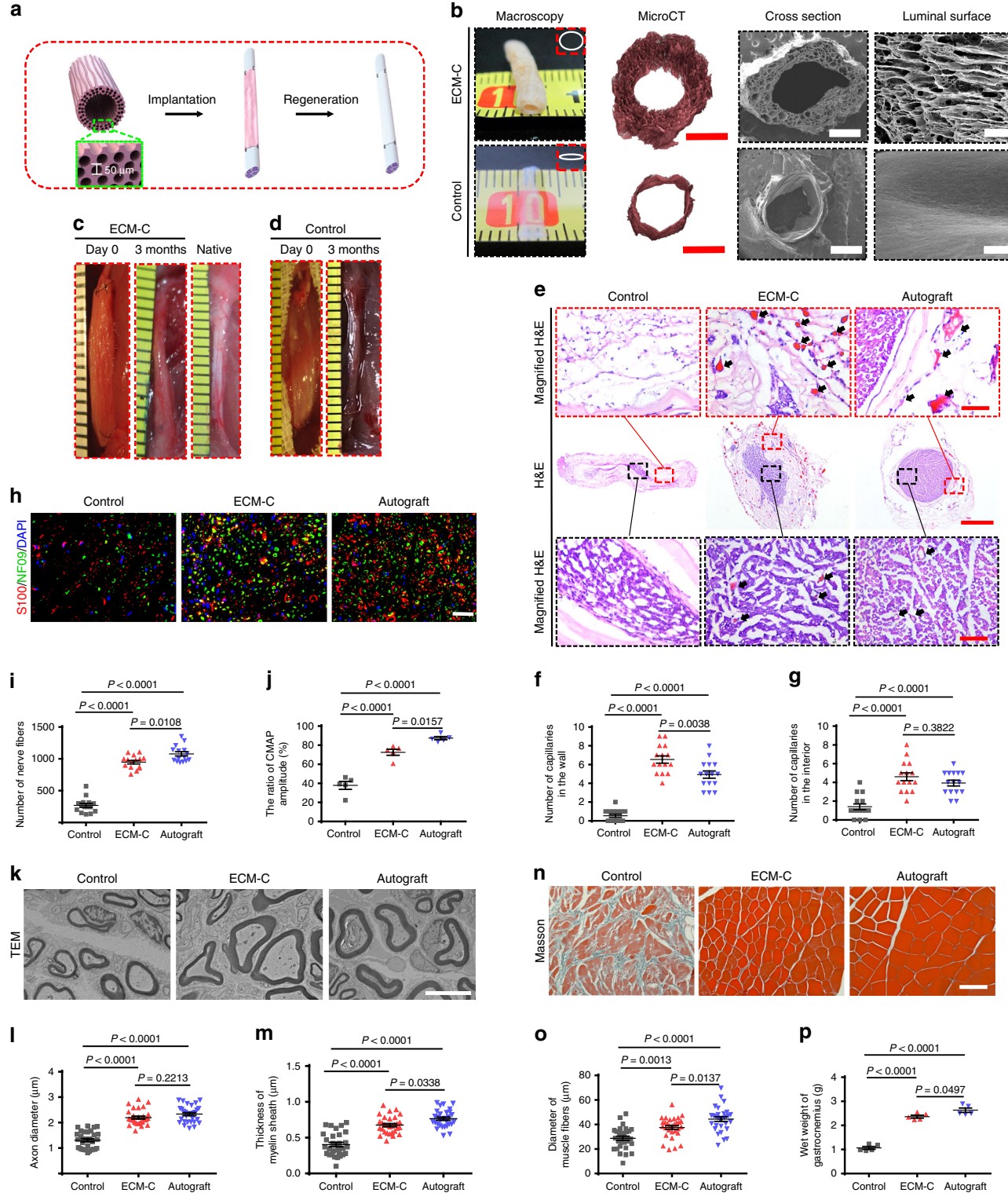

**Vascular regeneration**. To better induce the regeneration of arterial tissue with circumferential orientation for withstanding the pressure of the circulatory system, we therefore designed the tubular ECM-C with circumferentially oriented microchannels in the wall and evaluated in situ vascular regeneration and integration with abdominal artery defects (Fig. 7a). The inner diameter of the vascular ECM-C was 1.8 ± 0.1 mm, and the thickness of the wall was 423.2 ± 21.3 μm. The burst pressure of these scaffolds was 1489 ± 208 mmHg ($n = 4$), close to that of human

saphenous veins (1680 ± 307 mm Hg), the most commonly used autografts for coronary artery bypass surgery[35]. Good morphology preservation was observed with the vascular ECM-C (about 6 cm in length), i.e., no kinking occurred upon bending to 180° (Fig. 7b). MicroCT examination of such ECM-C revealed even distribution of circumferentially aligned microchannels within the wall. In contrast, control scaffolds (formed on the solid silicone rods) were only thin conduits without pores (Fig. 7c, Supplementary Fig. 8a, b, Supplementary Movie 5 and 6). SEM

**Fig. 6** Characterisation and utility of tubular scaffolds (ECM-C, control scaffolds and autografts) for nerve regeneration at 3 months. **a** Schematic illustration of implantation of tubular nerve ECM-C scaffolds for nerve regeneration of the rat sciatic nerve defects. **b** Macroscopic appearances and microstructures (MicroCT and SEM) of tubular scaffolds. Insets indicate the shape of obtained scaffolds. **c**, **d** Macroscopic view of the implanted (day 0) and regenerated nerves (3 months) by ECM-C and control scaffolds in rat sciatic nerve defects. **e** H&E staining of the transverse sections of regenerated nerves showing the distribution of regenerated capillaries and myelinated fibres in the scaffolds. Black arrows indicate the capillaries. Black and red boxed images are the magnified region of interior and wall of the explanted scaffolds. **f**, **g** Semi-quantification of capillaries in the interior and the wall of various scaffolds. **h** Immunofluorescence staining of the transverse sections of the explanted scaffolds confirmed that the myelinated fibres were positive for NF09 (green) and S100b (red). **i**, **j** The number of myelinated fibres and the CMAPs amplitude among various scaffolds ($n = 5$ animals per group). **k** Transmission electron micrographs of the transverse sections from various groups. **l**, **m** Comparison of the thickness of regenerated myelin sheath and diameter of regenerated myelinated fibres ($n = 5$ animals per group). **n** Masson trichrome staining of the cross-sections showing the morphology of gastrocnemius muscle. **o**, **p** Average diameter of muscle fibres and wet weight of gastrocnemius muscle ($n = 5$ animals per group). Bar heights and error bars represent means ± s.e.m. (ANOVA). Statistical analysis (ns = no significance). Scale bar: **b–d** macroscopic images, 1 mm; microCT images, 500 μm; SEM of cross section, 500 μm; SEM of luminal surface, 100 μm; **e** H&E, 200 μm; Magnified H&E, 50 μm; **h** 100 μm; **k** 5 μm; **n** 50 μm

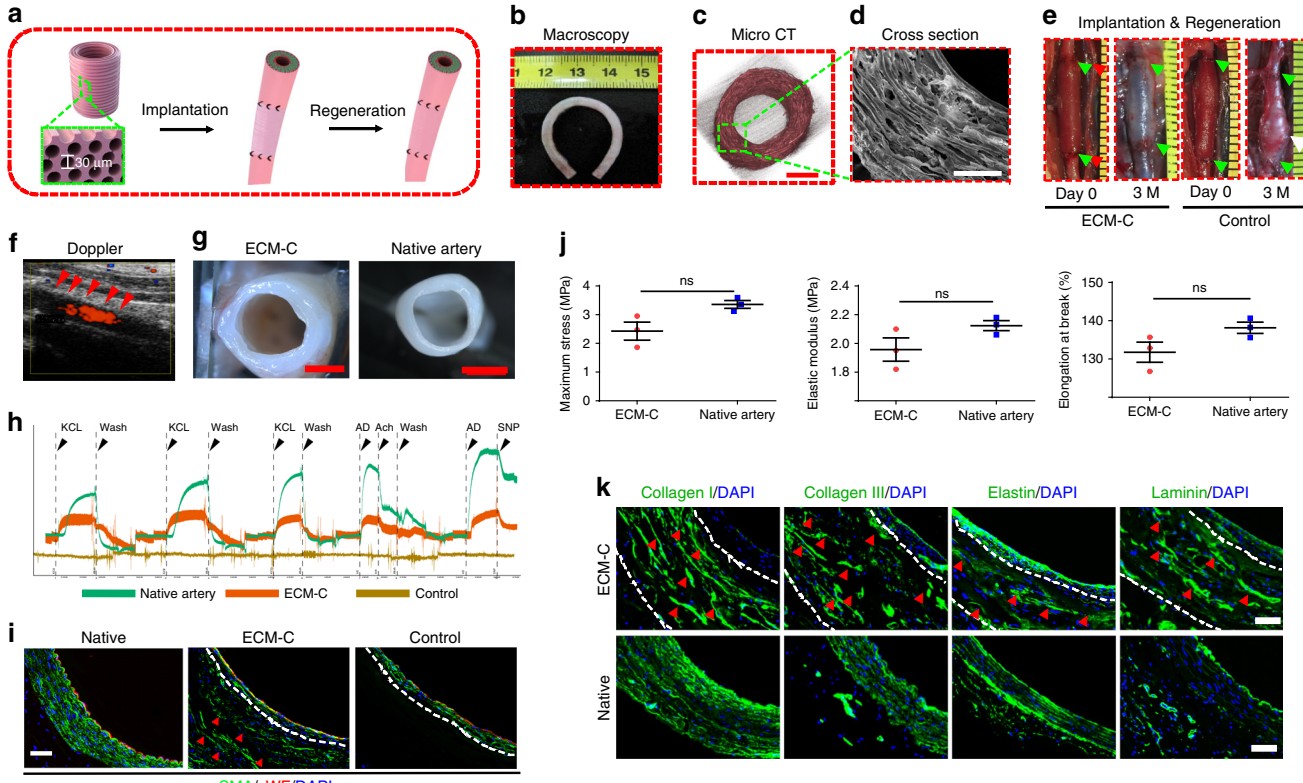

**Fig. 7** Characterisation and utility of tubular scaffolds with circumferentially aligned microchannels in vascular regeneration at 3 months. **a** Schematic illustration of implantation of the tubular vascular ECM-C scaffolds for artery regeneration of the rat abdominal artery defects. **b** No kinking formed when the tubular ECM-C scaffolds were folded at 180°. **c**, **d** Micro CT and SEM images showing the macroscopic and microscopic appearance of tubular ECM-C scaffolds with circumferentially aligned microchannels observed from the cross section. **e** The ECM-C vascular scaffolds can withstand the blood pressure after implantation and maintains for 3 months, and remodel into neo-artery, integrating with the host artery. The control scaffolds formed aneurysm at 3 months. The green arrowheads represent the suture line of the anastomosis. The white arrowheads represent the aneurysm formation site. **f** The patency was evaluated by Laser Doppler ultrasound imaging. The red arrowheads indicate the position of ECM-C guided neoartery. **g** The cross-sectional morphology of ECM-C guided neoartery was close to native artery under the stereomicroscope. **h** Examination of the physiological functions of neo-artery revealed obvious sensitivity to vasodilators and vasoconstrictors, which is similar to native artery. The functional signals of control scaffolds were undetectable ($n = 3$ animals per group). **i** Immunofluorescence double staining showed the distribution of endothelial cells (vWF, red) and vascular smooth muscle cells (α-SMA, green) in native artery, ECM-C guided neo-artery and control scaffolds. White dashed lines indicate the border of the scaffolds. Red arrowheads indicated the regenerated VSMC along the microchannels. **j** Comparison of the mechanical properties of neo-artery and native artery including maximum stress, elastic modulus, and elongation at break (from left to right) ($n = 3$ animals per group). **k** Immunofluorescence staining showing the distribution of collagen I and III, elastin and laminin in ECM-C guided neoartery and native artery. White dashed lines indicate the border of the ECM-C scaffolds. Red arrowheads represent the deposited ECM components within the microchannels. Bar heights and error bars represent means ± s.e.m. ($t$-test). Statistical analysis (ns = no significance). Scale bar: **c**, 500 μm; **d**, 200 μm; **g**, 1 mm; **i**, **k**, 50 μm

examination further confirmed the presence of the circumferentially aligned microchannels (an average diameter of 27.9 ± 5.2 µm) in the wall of vascular ECM-C (Fig. 7d). Implantation of tubular vascular ECM-C (1.1 cm in length) into the rat abdominal artery defects showed that they could immediately withstand the blood pressure as shown in Fig. 7e. After three months, neo-artery exhibited a similar appearance to native artery and was only distinguished by sutures in the anastomosis (Fig. 7e). As expected, aneurysm formation occurred to all the control scaffolds after 3 months of implantation (8/11), and 3 rats died of scaffolds rupture (3/11) (Fig. 7e). Laser Doppler ultrasound imaging of the neo-artery at 3 months demonstrated fast blood flow and high patency with all ECM-C (11/11) (Fig. 7f). More importantly, the neo-arteries were compliant and pulsed synchronously with the adjacent native artery as observed in doppler ultrasound and video, indicating complete integration with the host artery (Supplementary Movie 7 and 8). The regenerated media layer of neo-arteries also had the capacity to respond to vasomotor agonists, potassium chloride (KCL) and adrenaline (AD), despite that the magnitude of constriction was less than that of native artery (Fig. 7h, Supplementary Fig. 9a). ECM-C guided neo-artery also exhibited vasodilation to endothelium-independent vasodilators including endothelial-specific activators of acetylcholine (ACh) and sodium nitroprusside (SNP), implying the formation of cell–cell signalling capability between endothelial cells (ECs) and contractile vascular smooth muscle cells (VSMCs) (Fig. 7h, Supplementary Fig. 9a, b). Co-staining of ECs and VSMCs revealed that thick smooth muscle layers were separated from the blood by a continuous endothelium in the ECM-C guided neo-artery, similar to native artery (Fig. 7i). No clear response was seen with control scaffolds and only a thin neointima composed of several VSMC layers and a monolayer of ECs was located on the luminal side (Fig. 7i). H&E staining of both transverse and longitudinal cross-sections of ECM-C guided neo-arteries confirmed that the microchannels in the wall were occupied by vascular cells (Supplementary Fig. 10a). These cells were positive for α-SMA and smooth muscle myosin heavy chain (SM-MHC) protein, the phenotypic markers of contractile VSMCs in native arteries (Supplementary Fig. 10a, c). On the luminal surface of control scaffolds, a thin layer of α-SMA and SM-MHC positive cells were also identified (Supplementary Fig. 10b). Statistically, SM-MHC positive cells per view area in ECM-C guided neo-artery was significantly larger than that of control scaffolds, although both of them were obviously smaller than that of native artery (Supplementary Fig. 10d). Timely endothelialization of the luminal surface is essential to achieve the high patency. SEM examination of the luminal surfaces of both ECM-C guided neo-artery and native artery showed that the surface was clean and free of platelet aggregates or thrombi and was covered with elongated spindle-like ECs (Supplementary Fig. 11a, b). Enface staining of the luminal surface of ECM-C neo-artery and native artery confirmed complete coverage by CD31 positive cells with cobblestone-like morphology and oriented parallel to the direction of blood flow (Supplementary Fig. 11a, b). Immunofluorescence staining of the cross sections affirmed the presence of continuous monolayer of CD31- and CD144-positive ECs on the luminal surface of ECM-C guided artery and native artery (Supplementary Fig. 11a, b). Further characterisation of the mechanical properties (maximum stress, elastic modulus and elongation at break) of ECM-C guided neo-arteries and native arteries (Fig. 7j) showed no statistical differences in between. Immunofluorescence staining for key ECM components (collagen I, III, and elastin) indicated that substantial new ECM was secreted along the microchannels of ECM-C guided neo-artery and they organised in a circumferential fashion similar to that of native artery (Fig. 7k). More importantly, laminin, an

essential component of the vascular basement membrane, also distributed close to the luminal side of ECM-C guided neo-artery, similar to that in the native artery (Fig.7k). Despite great similarities in composition, structure and mechanical performance, noted difference still existed between ECM-C guided neo-artery and native artery.

## Discussion

Recruitment of endogenous cells for tissue formation, also called in situ tissue regeneration[4,6], represents a promising strategy to restore the lost tissues. Introduction of ECM-like scaffolds with unique inductive niches would favour the process[8,12]. Here, we developed an approach to fabricate ECM scaffolds with parallel aligned microchannels, which effectively regulated elongation, migration, proliferation and maturation of L6, RSC96 and A10 cells in vitro (Fig. 3), and promoted cellularization and vascularisation, as well as modulated inflammatory responses in vivo (Fig. 4), particularly for the regeneration of three representative oriented tissues (Figs. 5–7). The use of sacrificial templates composed of aligned microfibers to assist the formation of ECM-C within a subcutaneous tissue environment represents several unique advantages: (1) better guidance of host cell infiltration and neo-ECM organisation, (2) precise control of the pore size, shape and organisation via altering the configurations of the template for specific target tissues, (3) possibility of creating large volume ECM scaffolds, and (4) easy accessibility of subcutaneous location for implantation and explantation at a high success rate and (5) ready-to-use implants as allograft and heterograft upon decellularization.

PCL was particularly used as sacrificial templates for its demonstrated biocompatibility and slow degradation[36]. The superior biocompatibility of PCL would mitigate acute inflammatory response while slow degradation could better maintain the structural stability for host cells to infiltrate and deposit ECM during in vivo implantation. In addition, the low melting temperature of PCL offers a high processability to flexibly control fibre diameter and therefore, the size of microchannels of ECM-C[36]. Further, the use of 3D printed templates could readily achieve the structural complexity[25].

Cumulative evidences have demonstrated a close correlation between cell morphology and their phenotype[8]. Cell morphology on the residing substrate is controlled by the assembly of focal adhesions (i.e., size and distribution)[37], which then regulates the intracellular cytoskeletal dynamics and stress fibre reorganisation[38]. Through such cascades, the regulatory effect of substrate physical attributes on cell physiology is transduced[39]. In this study, two-tier regulatory effects by matrix could be identified: (1) initial tissue formation within the sacrificial template in subcutaneous pocket, and (2) tissue regeneration guided by ECM-C at the defect site. In the former, fibroblasts, aside from immune cells (macrophages and neutrophils)[40,41], are probably the major cells involving in deposition of neo-ECM and spatial organisation of PCL microfibers might influence the fibroblast responses (migration and ECM deposition). Previous studies did show that parallel microfibers could induce the alignment of newly synthesised ECM[19,42], which was consistent with our findings — new collagen fibrils (~50 nm in diameter) aligned in the same direction of PCL microfibers (Fig. 2f). Such aligned ECM would favour the myogenic phenotype of muscle cells or neurogenic phenotype of the nervous cells as a result of the topology guidance[43,44]. By using PCL microfibers of different diameters, the microchannel size of ECM-C could be well controlled, which subsequently influences tissue ingrowth. Early studies pointed out that the pore size of 50–160 µm could support the capillary ingrowth and larger than 140 µm became necessary for effective vascularisation of

large and dense tissues[45]. In the case of innervation, various pore sizes were reported for regenerating axon and neurite and a general consensus is that uniaxially oriented pores with the size of 20–80 μm favour neurite and axonal growth[44,46]. In recognition of the structural uniqueness of ECM-C, the first two immediate applications were therefore explored for repairing damaged skeletal-muscle tissue and sciatic nerve defects.

The use of 3D anisotropic scaffolds such as aligned porous scaffolds and micro/nano fibrous scaffolds[43,47] can encourage myogenic differentiation and myotube formation to regenerate functional muscle fibres. With this regard, a membranous template composed of parallel 150-μm PCL microfibers was used to generate the muscle ECM-C. The final microchannel size retained about 150 μm, slightly larger than that of the myotube. Thus, the combination of aligned ECM fibres and parallel microchannels in muscle ECM-C offers hierarchical instructive niches to guide the formation of myotubes and muscle fibres along with vascularisation and innervation (Fig. 5). Such results are actually consistent with early finding that decellularized muscle matrix with channelled networks from microvessels exhibited a better repair capacity than that without the network[21,22]. Apart from the topological conduciveness by the scaffolds, the so-elicited polarisation of macrophages (Fig. 4) would also contribute to muscle regeneration by recruiting endogenous stem cells via their secreted chemokines and cytokines[48].

Similar to muscle repair, suitable physical and guiding cues are essential to axonal growth in peripheral nerve repair[44,49]. The development of tubular nerve ECM-C with longitudinally aligned microchannels (~40 μm) offered noticeable guidance to Schwann cells (RSC96), and helped functional recovery of nerves and alleviation of the gastrocnemius muscle atrophy (Fig. 6). Once again, the encouraging results most likely result from the combinatory effect of oriented ECM nanofibers and parallel microchannels for enhanced cell migration and nutrient transport. In fact, a high number of capillaries distributed in the lumen and wall of the nerve ECM-C, which also ultimately impacted functional axon regeneration via mass and gas exchange[50]. Similarly, the contribution of macrophage polarisation (M2) to nerve regeneration cannot be neglected. However, the exact molecular mechanism to regulate axonal growth remains elusive[51,52].

Different from muscle and peripheral nerve, artery regeneration requires the formation of appropriate media layer to withstand the arterial pressure, coming from the circumferentially organised ECM fibres and smooth muscle cells[7,8]. Introduction of circumferentially parallel microchannels into the tubular ECM-C not only facilitated cell ingrowth, but also modulated cell phenotype (Fig. 7). In particular, the microchannels could guide the organisation of new ECM with great similarity to native artery (Fig. 7k). The unique combination of ECM-derived matrix and microchannels of vascular ECM-C determines their outperformance over those decellularized natural artery from human or animal tissues and in vitro constructed blood vessels, of which the compact structure may limit cells to efficiently infiltrate and subsequently regenerate physiological function[53–56]. Cumulative evidences have shown that microchannels could induce the orientation of VSMCs and promote the expression of contractile genes or proteins in vitro[57–59]. Our previous study demonstrated that the pore size of 30 μm in circumferentially aligned microfiber scaffolds could better support functional vascular regeneration in vivo, which serves as the basis for our design of the microchannel size for ECM-C. These microchannels indeed favoured the migration and orientation of VSMCs and their expression of contractile genes and proteins in vitro and in vivo for functional integration with host artery (Figs. 2 and 7). Furthermore, the circumferential orientation of new ECM in ECM-C guided neoartery could achieve the desired mechanical performance to

prevent the occurrence of aneurysms. The advantages of vascular ECM-C in guided vascular regeneration may be the combined result from phenotypic regulation of vascular cells and polarisation of macrophages[60,61].

The promising progress from clinical trials of in vivo engineered autologous ECM capsules[26] undoubtedly paves the potential avenue toward clinical translation of ECM-C. In light of the encouraging results with rats, we speculate that the demonstrated approach can be readily extended to large animal models. The possibility of engineering heterologous ECM scaffolds in humanised animals such as porcine holds even more potential to circumvent the need of human bioreactor, allowing for upscaling and mass production[62]. Adaption of the ECM scaffolds to cater for other specific tissue types beyond oriented tissues or as cell delivery vehicles to improve cell viability and engraftment requires specific design and fabrication of the sacrificial templates to accommodate the unique structural needs. Recruitment of specific cells into the sacrificial template via gene editing or cellular reprogramming can help produce tissue-specific ECM scaffolds and definitely warrants further exploration[63,64].

In conclusion, the present study demonstrated an accessible, safe and scalable approach to engineer off-the-shelf ECM scaffolds capable of guiding tissue regeneration and functional integration. In particular, the ability to tailor such ECM scaffolds (e.g., 3D macro/micro structure and topography) specifically for those target tissues for regeneration via the sacrificial templates opens a promising avenue toward personalised regenerative medicine. Furthermore, ECM scaffold-enabled in situ tissue regeneration, i.e., recruitment of endogenous cells and stimulation of self-healing process, represents an effective alternative to conventional tissue engineering, which remains to address the identified challenges, e.g., vascularisation, structural complexity, etc.

## Methods

**Animal experiments.** In all, 116 male Sprague Dawley rats (male, aged 8–10 weeks with the weight range of 280–320 g) were used in this study. All the animals were purchased from the Laboratory Animal Centre of the Academy of Military Medical Sciences (Beijing, China). Animal experiments were approved by the Animal Experiments Ethical Committee of Nankai University and complied with the Guide for Care and Use of Laboratory Animals. In total, 40 rats were used to fabricate membranous and tubular scaffolds; 10 rats were used to evaluate the cell infiltration, vascularisation and immunomodulatory properties of scaffolds, 66 rats were used for in situ implantation of muscle, nerve and vascular scaffolds. For all other experiments a minimum of $n = 5$ rats per group was set for obtaining numerical data.

**Materials.** Poly (ε-caprolactone) (PCL) pellets (Mn = 70,000–90,000 Da), ribonuclease (RNase) and deoxyribonuclease I (DNase I) were purchased from Sigma-Aldrich (St. Louis, Missouri, US). Sodium Dodecyl Sulfonate (SDS) was purchased from Alfa Aesar (London, England, UK). Analytical reagents including, chloroform, methanol, ethanol among others, were obtained from Tianjin Chemical Reagent Company (Tianjin, China).

**Extracellular matrix scaffold fabrication.** ECM scaffolds with aligned microchannels were fabricated according to the following process: First, aligned PCL microfiber meshes were used as templates, which were prepared by melt-spinning using house-made apparatus. PCL pellets were added into a 20-mL stainless steel syringe with a 20-G needle, which was placed above the collecting rods. The distance between the rods and the needle was set at 1.0 cm. The PCL pellets were heated to 100 °C for 1 h and melted until transparency before proceeding to melt spinning. Driven by a syringe pump, the spun fibres of PCL were extruded out at a flow rate of 2 mL per h and 0.5 mL per h for muscle and nerve scaffolds, respectively. For muscle scaffolds, the fibres were collected onto a rotating aluminium alloy rod (4.0 cm in diameter) at 50 rpm for 100 min. For nerve scaffolds, the templates were fabricated by overlaying PCL fibres onto the silicone rods (1.5 mm in diameter) and tightened with 6-0 suture at both ends. For artery scaffolds, the fibres were wrapped around silicone tubes (1.8 mm in diameter) encapsulating a stainless steel rod, providing a reliable fixation effect at 50 rpm for 8 min, with a slow flow rate of 0.5 ml per h. Second, cellularization of the PCL templates was achieved as follows: a small dorsal incision was made and the membranous fibre mats or the axially aligned fibres wrapped around silicon rods were implanted

subcutaneously in rats for 4 weeks. Third, the cell-ECM-PCL constructs were explanted and surrounding tissues were removed by trimming with scissors. The trimmed constructs were washed with sterilised saline water, and then sequentially dehydrated with 40%, 60%, 80 and 100% ethanol. The constructs were then immersed in chloroform for 48 h under gentle shaking to remove the PCL template, and chloroform was replaced every 12 h. The constructs were then sequentially rehydrated with 100%, 80%, 60%, and 40% ethanol until sterile purified water. Upon rehydration, the constructs were immersed in phosphate-buffered saline solution (PBS) containing 1% SDS followed by the treatment with 1% DNase I and RNase solution for 12 h at 37 °C to achieve decellularization. Porous ECM scaffolds with aligned microchannels were obtained after washing and freeze-drying using a Lab-1D-50 freeze-drier (Beijing Boyikang Laboratory Instruments Co., Ltd, Beijing, China). For control scaffolds, silicon membranes or rods coated with a thin layer of PCL (thickness = 12 μm) were implanted subcutaneously for 4 weeks, and following the similar aforementioned procedures to generate the decellularized control scaffolds. Thin PCL coating onto silicon membranes or rods was achieved by dipping into a solution of PCL in chloroform (0.1 g per mL) and dried at room temperature. Freeze-dried scaffolds were used for structural characterisation. The rest of scaffolds were sterilised by incubating in 75% ethanol for 1 h, and then stored in the sterile saline water at 4 °C until further use.

**Mechanical testing**. Mechanical properties of the sterilised scaffolds (maximum stress and strain at rupture) were measured on the Instron 3345 tensile-testing machine (Norwood, Massachusetts, USA) with a load capacity of 100 N. Scaffolds of 4.0 cm (L) × 1.0 cm (W) were clamped with a 1.0-cm inter-clamp distance and pulled longitudinally at a rate of 10 mm per min until rupture ($n = 5$). Scaffold elasticity was determined by calculating Young's modulus based on the slope of the stress–strain curve in the elastic region. Suture retention strength was measured after soaking in PBS at room temperature for 12 h. In brief, the suture (3-0; JinHuan, Shanghai, China) was inserted 2.0 mm from the edge of short axis of the sample, looped, and secured with three knots. One end of the sample was fixed to the stage clamp of the uniaxial load test machine (Instron-3345) and the opposite sutured end was fixed to another clamp. The distance between the clamps was set at 3.0 cm. The sample was pulled at a crosshead speed of 8 mm per min until rupture and suture retention strength was calculated. Burst pressure was measured with a customised instrument by filling the tubular ECM-C scaffold of 1.0 cm in length with petroleum-based lubricant, fastening one end and hermetically closing the other end to a vascular scaffold. A constant filling rate of 0.1 ml per min was applied, and the filling pressure was recorded until the rupture of the scaffold wall.

**MicroCT analysis**. Bruker SkyScan Micro-CT (SkyScan 1276, Allentown, PA, USA) was used for scanning. The scanning conditions were: 50 kv-70 mu A, no filter, with a spatial resolution of 3 μm. The obtained data were reconstructed by NRecon (Version: 1.7.1.6) software. Analysis of the region of interest (ROI) was selected by CTAn (Version: 1.17.9.0) software. Pore size, porosity, orientation and anisotropy were obtained, and 3D ROI map was reconstructed. CTvox (Version: 3.3.0.0) software was used for 3D image analysis and video production.

**Fibre and pore size measurements**. All the scaffolds were examined by using a FEI Quanta 200 scanning electron microscope (SEM; ThermoFisher Scientific Europe NanoPort, Eindhoven, Netherlands) at an accelerating voltage of 15 kV. Five randomly selected SEM images were acquired for each scaffold, and images (at least 30 fibres and 30 pores) were manually measured and analysed using ImageJ software v1.5 (NIH, Maryland, US) to calculate the average fibre diameter, microchannel diameter and pore size.

**Cell orientation, viability and proliferation on scaffolds in vitro**. The membranous scaffolds were cut into circular discs and used for cell culture. Rat skeletal muscle cell line L6, Schwann cell line RSC96 and Vascular smooth muscle cell line A10 (American Type Culture Collection, ATCC) were used to evaluate the regulating role of ECM-C scaffolds. RSC96 cells were cultured in 1640 Medium with 10% foetal bovine serum (FBS). L6 cells were cultured in medium containing 90% Dulbecco's Modified Eagle's Medium (DMEM) and 10% FBS. A10 cells were cultured in DMEM high glucose with 10% FBS. The cells cultured in the scaffolds for 1 or 3 days were used to observe cell morphology. Cytoskeleton organisation was visualised by fluorescently staining with phalloidin-AlexaFluor 488 (Sigma-Aldrich) and DAPI (Sigma-Aldrich). Images were taken by the laser scanning confocal microscope (Leica, Germany). The spreading area (S) and perimeter (L) of the nuclei at day 3 were measured with ImageJ software v1.5. The nuclear shape index (NSI) was calculated as ($4\pi S/L^2$). Nuclei with a linear and elongated morphology have NSI approaching 0. While nuclei with nearly circular shape have an NSI close to 1[42]. For cell migration assay, RSC96, A10 and L6 were labelled with a DiI Vybrant Solution (Invitrogen, USA) and then seeded on membranous scaffolds to evaluate cell migration. Live cell imaging was performed using the Infinity 3 2D array confocal scanner (Visitech Intl. Ltd. Norway) equipped with an Olympus IX81 inverted microscope system. The culture was maintained at 37 °C and 5% $CO_2$ throughout the process. Z stack images were acquired every 5 min for 4 h. Each individual cell was tracked using the "Manual Tracking" plug-in of NIH

ImageJ software to output (x, y) position coordinates during the observation period. The cells trajectories were rebuilt and the initial positions of all cells were automatically normalised to the original position (0, 0). The Euclidean distance (S) of cell migration was calculated by Chemotaxis and Migration Tool 2.0 (IBIDI, Germany) at 5-min intervals. The cell migration rate was calculated using the formula: v ¼ S/t (t ¼ 240 min). At least 40 cells were used for calculating the migration under each condition. Cell counting kit-8 (CCK-8) assay was used to detect the cell proliferation. Cell viability was analysed by live/dead (Invitrogen, American) staining after culture for 7 days.

**Real-time quantitative polymerase chain reaction analysis**. Real-time polymerase chain reaction (PCR) was used to investigate the regulatory effect of ECM-C and control scaffolds on gene expressions of L6, RSC96 and A10 cells. Total RNA was extracted from cell-seeded scaffolds at 7 days. RNA yield was determined by using a NanoDrop spectrophotometer (NanoDrop Technologies). PCR was executed on a CFX96 Real-Time PCR System (Bio-Rad, Hercules, CA) with a SYBR Green based real-time detection method (Roche, Mannheim, Germany). The expression of *Myh* (myosin heavy chain), *Myog* (myogenin) of L6 cells, and *Nrcam* (neuronal cellular adhesion molecules), *Krox20* (early growth response 2) and *Ngf* (nerve growth factor) of RSC96 cells, *Acta2* (α-smooth muscle actin) and *Myh11* (smooth muscle myosin heavy chain) of A10 cells and the housekeeping gene glyceraldehyde phosphate dehydrogenase (GAPDH) of corresponding cells were detected. Relative expression level of corresponding mRNA was expressed as $2^{-(\Delta\Delta CT)}$ method and normalised by housekeeping gene GAPDH. The primer sequences are listed in Supplementary Table 1.

**Characterisation of cell infiltration, vascularisation and immune response**. To evaluate the cellularization, vascularisation and immunoregulatory capacity, the control and ECM-C scaffolds were subcutaneously re-implanted into rats for 2 weeks. DAPI staining was used to evaluate the cell infiltration; H&E staining and immunofluorescence staining for von Willebrand factor (vWF) were used to assess vascularisation. To identify macrophage phenotype, immunofluorescence staining was performed with the following antibodies: CD68 (1:100, Abcam, ab31630), a general macrophage marker; CD206 (1:200, Abcam, ab64693), a marker for M2-like macrophages and inducible nitric oxide synthase (iNOS) (1:200, Abcam, ab15323) as a marker for M1-like macrophages. For immunofluorescence staining, frozen sections were incubated with 5% normal goat serum (Zhongshan Golden bridge Biotechnology, China) for 30 min at room temperature. For intracellular antigen staining, 0.1% Triton-PBS was used to permeate the membrane before incubation with serum. Then the sections were incubated with primary antibodies in PBS overnight at 4 °C, followed by incubation with secondary antibody in PBS for 2 h at room temperature.

**In vivo implantation of muscle scaffolds and corresponding evaluation**. For the implantation of muscle ECM-C and control scaffolds, the left hind limb was shaved after the rats were anaesthetised, and the incision was made in hind limb to expose the tibialis anterior muscle. The volume of muscle with the dimensions of 1.0 cm × 0.8 cm × 0.5 cm (length × width × height) was removed with a scalpel, and the removal weight was about 25% of the total weight of tibialis anterior muscle. In all, 3–5 pieces membranous ECM-C or control scaffolds of a similar size to muscle defect were then immediately transplanted into the defect site of the recipient animals. Six to eight stitches were required to immobilise the scaffolds. Upon closure of the skin by suturing, and the wound was cleaned with saline solution. After 4 weeks, CMAPs was obtained from an electrode located in the centre of the tibialis anterior muscle belly and a reference electrode positioned in the connective tissue of the knee joint. Finally, the regenerated muscle was harvested for histologic analyses.

**In vivo implantation of nerve scaffolds and corresponding evaluation**. For the implantation of nerve ECM-C and control scaffolds (diameter=1.5 mm; length=15.0 mm), the left sciatic nerve of rats was exposed carefully by blunt splitting of underlying muscle in the left lateral thigh. A segment of sciatic nerve was removed, leaving a defect of same length after retraction of the nerve stumps, and then sutured with 9–0 nylon sutures. In the autograft group, the transected nerve segment was sutured back under the microscope. After surgery, the rats were allowed to recover for 12 weeks. For characterisation, rats of each group were firstly assessed electrophysiologically, and then the explants were harvested for immunofluorescence staining. The CMAPs were recorded at the ipsilateral side by an 8-channel physiological signal recorder (RM-6280C, Chengdu Instrument Factory, Chengdu, China), and the peak amplitude of CMAPs was compared among groups. Regenerated nerves were harvested, fixed, and sectioned after recording the CMAPs. The thin sections were stained with S100 (1:200, Abcam; ab52642) and NF09 antibodies (1:200, Abcam; ab7794). Morphometric analysis was performed as follows: transverse ultrathin (thickness: 50.0 nm) sections were cut from the middle part of the regenerated nerves and stained with uranyl acetate and lead citrate, and then examined under a transmission electron microscope (HITACHI H-600, Hitachi, Japan). Images were taken from five randomly selected fields of each sample to determine the diameter of myelinated nerve fibres and the thickness of myelin sheaths with Image J software v1.5. The gastrocnemius muscles of both

hind limbs for each group were harvested, weighed, fixed, and sectioned and were assessed after Masson's trichrome staining. The average diameter of muscle fibres was obtained from at least 30 for each sample with Image J software v1.5.

**In vivo implantation of vascular scaffolds and corresponding evaluation**. For implantation of vascular ECM-C and control scaffolds, heparin (100 unit per kg) was used for anticoagulation by tail vein injection prior to surgery. A midline laparotomy incision was made, and the abdominal aorta was isolated, clamped, and transected. The tubular ECM-C scaffolds were sewed in an end-to-end fashion with 8 interrupted stitches using 9-0 monofilament nylon sutures (Yuan Hong, Shanghai, China). The wound was closed with 3-0 monofilament nylon sutures. No anticoagulation drug was administered after surgery. At 3 months, the patency of the implanted scaffolds was visualised using the high-resolution ultrasound (Vevo 2100 System, Visualsonics, Canada) after the rats were anaesthetised with iso-flurane. Vascular scaffolds were explanted under anaesthetisation. Aortic ring bioassay was performed according to the instrument standard operating procedure. Isometric forces were recorded with force transducers connected to a PowerLab/ 870 Eight-channel 100kHz A/D converter (AD Instruments, Sydney, Australia)[42]. Results were obtained from three individual rings derived from three explanted scaffolds.

For SEM analysis, the explants were rinsed with PBS and fixed with 2.5% glutaraldehyde overnight at 4 °C and dehydrated in ascending series of ethanol. Samples were sputter-coated with gold and observed by SEM (Quanta200, Czech Republic). For sectioning and staining, the explants were fixed with 4% paraformaldehyde for 1 h, dehydrated by 30% sucrose solution until the explants sank to the bottom. The explants were sectioned to 6-μm-thick slices after embedded in OCT. Then, the sections were stained with haematoxylin and eosin (H&E). Images were observed under the upright microscope (Leica DM3000, Germany). For immunofluorescence staining, the frozen sections were rinsed once with 150 mM PBS. The slides were then incubated in 5% normal goat serum (Zhongshan Golden Bridge Biotechnology) for 30 min at room temperature. For intracellular antigen staining, the cell membrane was permeated with 0.1% Triton-PBS and then incubated with primary antibodies in PBS overnight at 4 °C, followed by incubation with secondary antibody in PBS for 2 h at room temperature.

Endothelial cell staining was performed using rabbit anti-von Willebrand factor (1:200, Dako, A0082), mouse anti-CD31 (1:70, Abcam; ab64543) and CD144 primary antibody (1:100, Santa Cruz Biotechnology, sc-6458). The smooth muscle cells were stained using mouse anti-α-SMA (1:100, Abcam; ab7817) and mouse anti-smooth muscle myosin heavy chain I (1:100, Santa Cruz Biotechnology; sc-6956) primary antibody. ECM staining was performed using rabbit polyclonal anti-collagen I primary antibody (1:200, Abcam; ab34710), rabbit polyclonal anti-collagen III primary antibody (1:200, Abcam; ab7778), rabbit polyclonal anti-elastin (1:200, Abcam; ab21610) primary antibody, anti-laminin primary antibody (1:200, Abcam; ab11575) for collagen I, collagen III, elastin, laminin respectively. Alexa Fluor 488 goat anti-rabbit IgG (1:200, Invitrogen,) and Alexa Fluor 594 goat anti-mouse IgG (1:200, Invitrogen) were used as the secondary antibodies, respectively. The sections incubated with PBS without primary antibodies were used as negative controls. Slides were examined under a fluorescence microscope (Zeiss Axio Imager Z1, Germany).

**Statistical analysis**. All quantitative results were obtained from at least three samples and from three independent experiments. Data were presented as means ± s.e.m. All statistical analyses were performed in Graphpad Prism 7. Single comparison was made with an unpaired Student's $t$-test. Multiple comparisons were carried out using one-way analysis of variance (ANOVA) and Tukey's post-hoc analysis.

**Reporting summary**. Further information on research design is available in the Nature Research Reporting Summary linked to this article.

## Data availability
The data in this work are available in the manuscript or Supplementary Information, or available from the corresponding author upon reasonable request.

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

## Acknowledgements

This work was supported by the National Key Research and Development Program of China (2017YFC1103500), Innovative Research Group Project (81921004), National Natural Science Foundation of China (NSFC) projects (81972063, 81530059, 81772000 and 81601625), NSFC Fellowship Fund for International Young Scientists (81850410552), Science and Technology Support Program of Tianjin (16YFZCSY01020), National Science Foundation (NSF-DMR award number 1508511) and NIAMS award number 1R01AR067859 and the China Postdoctoral Science Foundation (#2016M590197).

## Author contributions

M.Z. and D.K. conceived the research; M.Z., D.K., H.J.W. and P.X.M. designed the experiments; M.Z., W.L., H.C., X.D. X.Y.Y. and K.W. performed the experiments; M.Z., W.L., H.C. and X.D. designed, fabricated and characterised scaffolds; M.Z., W.L. and K.W. performed microsurgery of animal experiments. M.Z., H.J.W., A.C.M., Y.H.W., H.Y.W, P.X.M., K.W. and D.K. interpreted the data, analysed the data and wrote the manuscript. All authors discussed the data and direction of the project at regular intervals throughout the study.

## Competing interests

The authors declare no competing interests.
