## [Peer Review File · Nature Communications]

Reviewers' comments:

Reviewer #1 (Remarks to the Author):

In the present study the authors engineered extracellular matrix (ECM) scaffolds with specific template to facilitate cellular engraftment and functional alignment. Overall the article is interesting and present good-looking data on the in vivo use of aligned porous scaffolds, for muscle, vasculature and nerve regeneration. The idea of using porous scaffolds to facilitate cell migration and of using the foreign-body response, to generate ECM scaffolds are not novel but the combination of the two retain a certain level of novelty. Moreover, while the approach of generating and engineering decellularised scaffolds for transplantation has reached clinical trial for vessels (Niklason et al), other tissues such as nerve and muscle benefits of pre-clinical trial. My general feeling is that the paper is interesting but the experiments have been done in a superficial way. There is a clear inconsistency in the controls that have been used across the paper and the mostly used control material is not ideal. The Bovine Pericardium is not the current gold standard, as it's known that the cross-linked nature of this scaffold significantly impact on the migration of cells into it. In contrast, other products such as the SIS have undergone extensive research and have also been used clinically for example for Volumetric Muscle Loss (Badylak et al; although the results presented are still debatable).

Specific comments on the paper:

Introduction:

In general, the introduction is too general and specific influence of designed ECM for the tissues of interest should be taken into account and discussed here.

Specifically:

Page 3: I would tone down the following part and specify here that this is true only for some specific tissues: "ECM scaffolds derived from allogeneic or xenogeneic decellularized tissues and organs have been shown to elicit a variety of favourable bioactivity and cell responses¹¹⁻¹³, nevertheless, the dense micro-architecture and constrained geometries of mature ECM can inhibit endogenous cell infiltration and thus, result in limited tissue remodelling and functional integration^{7, 14-16}."

Page 4: Again too general. This only applies to some tissues.

"Similar problems also exist when using other methods to generate decellularized ECM (dECM) scaffolds derived from native tissues¹⁴."

Results

Page 8, last sentence shows tense inconsistency (HAD – IS)

Page 8 I would compare the scaffolds with the tissues or decellularised tissues that they claim to use the scaffold for (nerve, artery, muscle), not just with bovine pericardium "There was no significant differences between the maximum stress, elastic modulus or suture strength of the ECM scaffolds and the bovine pericardium (Fig. 2g)."

Page 8 I would like to see an h&e and a mechanical test after 3 months

The ECM scaffolds were sterilized with 75% ethanol for 12 hours, and then were stored in saline water at 4°C for more than 3 months until it's time to use it.

Page 10, paragraph 2 At 1 day culture period is insufficient to evaluate cell migration, alignment and distribution. A more extended experiments needs to be performed, with different time-points and conditions. This will need to be included somehow in the main figures (maybe as part of Fig1)

to give the reader a clear immediate view of the results. The evaluation of nuclear shape can be insufficient to indicate the circularity of the cells, at variance with i.e. the measurement of the cell shape and size through a cytoplasmic staining. More details on the technique utilized should be provided in the methods section.

Fig 2G, It's not clear if the force measurements have been performed before or after the EtOH sterilization, this needs to be specified as the process is known to alter the mechanical properties of the scaffolds.

Page 11 The authors needs to specify the reason for choosing the Bovine Pericardium as a control scaffold. The article aims at repairing skeletal muscle, vascular and nerve defects and with the availability of decellularized scaffolds generated from every of these tissues, these could have been a better matched control. The total absence of capillaries within the pericardium after implantation, confirms that the scaffold is not an ideal control to assess cell migration. Similarly, in my opinion this bias the observations reported in Figure 3.

Fig 3: Here is a bit confusing il4 and il10 have a signature which is more anti-inflammatory or at least immune-modulating. I was expecting higher values of these two for their scaffold. In particular il4, which is definitely an m2 associated cytokine.
"...IL-4 (i), IL-10 (j)..."

Fig 4 The figure lacks the most basic controls. It shows the comparison of an untreated defects, with treated ones, in which any treatment would have in any case produced a better outcome than leaving an empty gap. The deposition of large amount of ECM components highlighted by the authors in figure 4F may indicate a fibrotic response. Please explain.

The volume recovery reported in 4G is compared to an unrepaired damage, therefore presenting a misleading result to the reader. The quantification reported in 4H indicates the % of fibers / area, this panel should also include the quantification of the number of fibers present in the area. The number of neuromuscular junctions / area presented in 4I and S5a require quantification as the author speculate on the meaning of the numbers detected in the control group. Overall the lack of appropriate controls makes these results severely incomplete. I suggest the authors repeat the experiments using a ECM control (i.e. pericardium or even better SIS), and quantify the results in comparison to their scaffold.

Fig S8 The figure seem to be flipped, with native on the top (a-e) and neo-artery in (f-j). If this is not the case, the native vessel presented in I and J seems otherwise to be completely "de-endothelialized". I feel that there is something wrong here.

Figure 5 The red VWF staining presented in G is not visible, please present split-channel images as space is available in the figure.

Figure 6 Overall I think that the study is a bit confusing, the three experiments are performed with three different control setups, making a bit difficult to have an overall comparison, I would prefer to see the same controls in every experiment, it could be that they have the right controls, but they showed only the ones who makes their scaffold more appealing.

It's not clear to me how the nerve regeneration was assessed. The staining presented in 6F is not clear, looks more background and the recognition of the axons is hard to the sight. A higher magnification and an un-injured nerve staining will be required here.

It is difficult to understand how the number of regenerated axons visible in red in 6L could not possible sustain the level of regeneration presented in 6O.

Discussion

Page 30 The authors define the peritoneal cavity "a bioreactor". It would use more broad terms such as "bioreactor-like site" or "scaffold-remodeling site".

This section should be toned down or modified, the current results of decellularized materials are better than what is described here: "However, bovine pericardium, a popular material choice for use as a filler or to provide mechanical support for tissue defects, exhibited limited infiltration of host cells due to its dense pore structure. Indeed, a large number of reports have demonstrated that acellular matrices, such as small intestinal submucosa (SIS), urinary bladder, arteries, heart valves, the fascia lata, the dermis and tendon failed to support host cell infiltration, and led to inadequate tissue remodelling and functional integration¹⁴, 38-43. "

Page 33 The authors speculate on the use of their scaffolds in large animals and clinical studies. It needs to be mentioned in the discussion the problem of gas and nutrient diffusion in large volume of tissue / scaffolds. The last sentence of the page on reprogramming and gene editing is not clear, looks out of context and lacks of references.

This sentence is true, but the interleukins profile seems in contrast with this, please explain: "Compared with the bovine pericardium, ECM scaffolds with aligned microchannels favoured M2 macrophage phenotypic switching."

Reference:

22 and 34 is the same article

Overall, the paper is interesting but confusing at time. The three experiments are performed with three different control setups, making a bit difficult to have an overall comparison; I am not sure bovine pericardium is the most appropriate control.

I think that they don't properly comment the potential clinical translation of this approach, which, to me, seems at least a bit complicate.

Reviewer #2 (Remarks to the Author):

The authors describe the development and use of a novel scaffold derived from ECM but engineered to contain a "porous" channel network that allowed for greater cell in growth and organized alignment when used as a vascular, muscle and nerve template. Overall the experimental design is thorough, and it is commendable that the authors have chosen to demonstrate function in three distinct animal models to demonstrate functionality. Having said that, there are several points that the authors should address and clarify.

1. The authors talk in the introduction about porosity and the lack of pores in decellularized ECM materials and the need to increase porosity to support better remodeling. This is a very limited view and is certainly an engineer's perspective and not a biologist's perspective. When in place in the body as part of normal tissue the ECM has no pores, the pores that are observed by microscopy as generally an artifact of the scaffold being hydrated, normal tissues are not full of holes, those holes are filled with hydrated proteoglycans and other proteins that cells have to push through, there are no channels of pores in normal tissues and ECM. I would argue that in this study the authors are not developing a scaffold with increased porosity but have simply engineered and ECM scaffold to have aligned channels to promote cell migration, that is to say the ECM that forms around the template scaffold has a "porosity" that is independent of anything that authors are able to engineer. The leaching of the template scaffold simply creates channels in the matrix that act as conduits for cells the "porosity" of the actual ECM, if you could call it that, is unaffected. I would suggest that the authors consider revising their use of the term porosity and consider a better description for their hybrid scaffold as generating conduit channels rather than pores.

2. In the section on the immunomodulatory effects of the scaffolds I do not fully understand the

choice of scaffolds and how the macrophage polarization results compare. The authors state that they used autologous, allogeneic and heterologous ECM scaffolds. It is not stated what how the autologous ECM scaffold was prepared, given that the production of the scaffold was 4-weeks not including the 7-8 day decellularization process. Can the authors be more definitive that this was decellularized ECM derived from the same animal? The allogeneic scaffolds and the heterogeneic scaffolds were also compared to bovine pericardium as a control. Given that this is another example of a heterogeneic scaffold I feel the authors need to give more details as to why this particular scaffold acted as control. Was this based on ability to compare to prior data i.e. to confirm an appropriate host response was occurring or was this a true negative control that the authors knew would generate an unfavorable host response to which the others could be compared? There is a wealth of data already published demonstrating that xenogeneic and allogeneic ECM prepared from decellularized tissues does not elicit a foreign body reaction or any kind of immunological rejection response, so it seems excessive to perform these experiments on so many different ECM sources without discussing the relevance of the results in more detail.

3. One would argue that soaking in 75% ethanol for 12 hours is not a sterilization technique, but a disinfection technique, the authors do not demonstrate that the scaffolds meet sterility standards following this step.
4. As previously state the use of ECM scaffolds and decellularized tissues for regenerative medicine is well established and there are a plethora of examples of their success, regardless of their lack of "pores" or microchannels. I feel it would be helpful in the authors could make a more definitive statement about why there is a need for engineered decellularized scaffolds. Can the authors provide examples where ECM scaffolds have failed as a result of lack of cell alignment or cellular infiltration? Or provide more specific descriptions of injuries that required an engineered scaffold or where microchannels would be necessary. I can see how microchannels would be beneficial for nerve conduits, but additional examples would be helpful for the reader.
5. The authors report that "Previous studies demonstrated that polymer-based matrices could be used as pore-forming templates to generate porous ECM scaffolds in vitro^{17, 30}, however, these scaffolds possessing low mechanical strength restrict their in vivo implantation." The authors, however, do not report the mechanical properties of these competitor scaffolds, nor do they report the mechanical strength that would be required for any of the native tissues into which they implanted their scaffolds. Without this information, it is very difficult to assess whether the mechanical properties of these new scaffolds are sufficient. Further, the authors (in Figure 2) only show the mechanical properties of one of their prepared scaffolds, but they report having generated three different types of scaffolds, each of which would ostensibly have unique mechanical properties. The authors should perform mechanical testing on all of their materials or explicitly state why they chose not to do so.
6. Similarly, the choice of control in their experiments may not be appropriate. The authors used bovine pericardium because it is a commonly used ECM product. This material, however, is not derived from the same species or the same anatomic location as the ECM from which the authors are generating their scaffolds. It would be ideal to utilize ECM derived from a subcutaneous pocket into which a sham material was placed, or no material was placed. In this way, comparisons between porosity, mechanical properties and cellular behavior can be made more readily and significant changes in cellular behavior can be directly attributed to the use of a sacrificial PCL scaffold, rather than differences in tissue type. Furthermore, a native sample should be included in Figure 2f.
7. On page 8 of the results section regarding cell alignment and cytocompatibility in vitro, the choice of control is again problematic. As a control, the authors use collagen scaffolds. This seems inappropriate because there are myriad other components in an ECM scaffold than just collagen, most of which will have bioactivity. It is more appropriate to compare any results against another ECM type such as dermis or urinary bladder ECM.
8. It is unclear why the authors chose to generate their porous ECM scaffolds in different animal types (rat vs. rabbit), and at no point in the manuscript does this objective arise prior to pg. 8. The authors need to more thoroughly introduce this reasoning.
9. The functional test used to determine functional muscle tissue deposition, the jump test, is somewhat qualitative. Furthermore, 30 days is not a long enough period of time for mature muscle

fiber deposition, with 60 days being a more appropriate timepoint. More rigorous tests would have included isometric torque analysis plus EMG. The authors should ideally perform these studies, as well, or at least acknowledge the limited scope of their chosen functional tests.

10. Within the same section, pg. 13, the authors state that “decellularized muscle tissue scaffolds were also implanted subcutaneously, showing few cell infiltration and vascularization with sparse distribution of an amount of CD206 positive cells and some capillaries around the border of the scaffolds.” The reviewer is left asking why this study was done, what “decellularized muscle tissue scaffolds” represents, and what controls were performed in tandem with this material? Greater detail is required when discussing this experiment and its results, as well as a more uniform system of reference for the materials used throughout this manuscript.

11. For the vascular results, the results section should include which arteries these scaffolds were implanted into, and what the patency of the ECM scaffolds is being compared to (are they compared to native controls?). Furthermore, the results showing that patency was nearly 100% is surprising and should be placed in line with previously reported results. These results are not addressed later in the discussion.

12. For the nerve resection, the size of the defect should be included in the results section.

13. Any time a significant difference is noted, the p-value needs to be reported.

14. In the methods section, more detail is needed regarding antibodies used for immunostaining. In particular, the composition of blocking antibodies, the duration of incubation with blocking buffer, and the dilutions of antibodies are all necessary.

15. “polylactide acid (PLA)” should be changed for “Polylactic acid (PLA)”

16. In the sentence “have been reported to induce acute, and in some cases,” change “is for “in”

17. In the phrase “Biocompatible and porous ECM scaffolds” correct the underlined word for “Biocompatible”

18. The authors claim that “no significant difference was found between the heterologous ECM and pericardium scaffolds (Fig. 3f)”, however, the graph that compares those 2 groups has an * ($p < 0.05$) indicating significant differences. This should be corrected.

19. “CMAP” is mentioned for the first time in the caption of figure 4, but the acronym description occurs later in the paper: “The ratio of the compound muscle action potentials (CMAP)...”

20. In some cases, there are 2 spaces between words.

Reviewer #3 (Remarks to the Author):

The authors designed an extracellular matrix-based bioscaffold with good biocompatibility and low immune response and examined the in vivo effect of the novel bioscaffold. Their outcomes powerfully showed that the novel bioscaffold could guide cell alignment, promote cellularization, encourage vascularization, regulate macrophage M1/M2 state switch. Functional outcomes demonstrated that the bioscaffold was able to promote the regeneration of skeletal muscle, arterial tissue, and nerves.

The study is integrated and well-designed. Statistical methods are suitably used as well. One minor comment is that the authors should study or at least discuss more about the underlying cellular and mechanisms of the promoting role of extracellular matrix-based bioscaffold in tissue engineering. I would suggest “accept with minor revision”.

Point-to-point response to reviewers' comments:

Reviewer #1:

Comment #1: In the present study the authors engineered extracellular matrix (ECM) scaffolds with specific template to facilitate cellular engraftment and functional alignment. Overall the article is interesting and present good-looking data on the in vivo use of aligned porous scaffolds, for muscle, vasculature and nerve regeneration. The idea of using porous scaffolds to facilitate cell migration and of using the foreign-body response, to generate ECM scaffolds are not novel but the combination of the two retain a certain level of novelty. Moreover, while the approach of generating and engineering decellularised scaffolds for transplantation has reached clinical trial for vessels (Niklason et al), other tissues such as nerve and muscle benefits of pre-clinical trial. My general feeling is that the paper is interesting but the experiments have been done in a superficial way.

Response: First of all, we are grateful to the reviewer's insightful comments on the decellularized matrix for tissue regeneration while recognizing the novelty of our manuscript. The key innovation of our study rests on that sacrificial template-assisted in vivo engineering of ECM-derived scaffolds with parallel orientated microchannels (ECM-C), which can guide cellular morphology, improve cell migration, and promote the regeneration of orientated tissues. The introduction of microchannels into ECM-derived scaffolds becomes essential for tissue ingrowth and integration with host tissue to restore the physiologic functions. This is different from the efforts from Niklason et al, in which ECM-based vascular grafts with dense structure were generated using the in vitro culture, and no additional porous structure was created to recruit endogenous cells for promoted vascular regeneration, even in the clinical hemodialysis conduit experiments [1-3]. Following the comments, we have performed quite a few additional in vitro and in vivo experiments to further elaborate the novelty and importance of the well-designed scaffolds for oriented tissue regeneration together with our initial findings. We have also made additional efforts to address and discuss the novelty and importance in the revised manuscript.

1. ZH, S. et al. A completely biological "off-the-shelf" arteriovenous graft that recellularizes in baboons. Science translational medicine 9, (414):10.1126/scitranslmed.aan4209 (2017).
2. RD, K. et al. Bioengineered human acellular vessels recellularize and evolve into living blood vessels after human implantation. Science translational medicine 11,(485): DOI: 10.1126/scitranslmed.aau6934; (2019).
3. SL, D. et al. Readily available tissue-engineered vascular grafts. Science translational medicine 3, 68-69 (2011).

Comment#2: There is a clear inconsistency in the controls that have been used across the paper and the mostly used control material is not ideal. The Bovine Pericardium is not the current gold standard, as it's known that the cross-linked nature of this scaffold significantly impacts on the migration of cells into it. In contrast, other

products such as the SIS have undergone extensive research and have also been used clinically for example for Volumetric Muscle Loss (Badylak et al; although the results presented are still debatable).

Response: Again, we are greatly thankful to the constructive comments on the controls. Combined with the similar advice from Reviewer 2, we have redesigned the controls across all the study and the appropriate controls would be the decellularized *in vivo* constructed-ECM scaffolds without microchannels. With such controls, the importance of microchannels in oriented tissue regeneration could be better appreciated. To generate control scaffolds, silicon membranes or rods coated with a thin layer of PCL (thickness= about 12 μ m) were implanted subcutaneously for 4 weeks, and followed by the removal of polymer and cells to generate the decellularized control scaffolds. Thin PCL coating onto silicon membranes (for muscle repair) or rods (for nerve repair) was achieved by dipping into a solution of PCL in chloroform (0.1g/mL) and vacuum dried at room temperature. The rationale and innovation were largely embodied in the revision following the comments.

Specific comments on the paper:

Introduction:

Comment #3: In general, the introduction is too general and specific influence of designed ECM for the tissues of interest should be taken into account and discussed here.

Response: Taking the advice, we have revised the introduction completely with specific emphasis on the design of ECM scaffolds with oriented microchannels for oriented tissue regeneration, please see the revised introduction.

Specifically:

Comment #4: Page 3: I would tone down the following part and specify here that this is true only for some specific tissues: “ECM scaffolds derived from allogeneic or xenogeneic decellularized tissues and organs have been shown to elicit a variety of favourable bioactivity and cell responses¹¹⁻¹³, nevertheless, the dense micro-architecture and constrained geometries of mature ECM can inhibit endogenous cell infiltration and thus, result in limited tissue remodelling and functional integration^{7, 14-16}.”

Response: Thanks for the specific suggestion. We have revised accordingly in the introduction section.

Comment #5: Page 4: Again too general. This only applies to some tissues.

“Similar problems also exist when using other methods to generate decellularized ECM (dECM) scaffolds derived from native tissues¹⁴.”

Response: We have revised it accordingly as well.

Results

Comment #6: Page 8, last sentence shows tense inconsistency (HAD – IS)

Response: Sorry for the inconsistent tense in the context and the manuscript was

revised with additional editorial assistance.

Comment #7: Page 8 I would compare the scaffolds with the tissues or decellularised tissues that they claim to use the scaffold for (nerve, artery, muscle), not just with bovine pericardium “There was no significant differences between the maximum stress, elastic modulus or suture strength of the ECM scaffolds and the bovine pericardium (Fig. 2g).”

Response: Thanks for the very valuable comments. As mentioned in Comment #2, we redesigned the controls to match for each application. The mechanical properties of the scaffolds were also characterized as shown in Fig 4S.

Comment #8: Page 8 I would like to see an h&e and a mechanical test after 3 months The ECM scaffolds were sterilized with 75% ethanol for 12 hours, and then were stored in saline water at 4°C for more than 3 months until it's time to use it.

Response: Following the recommendation, both H&E staining and mechanical testing were conducted to the scaffolds. H&E staining showed that the arrangement and density of ECM in the prepared scaffolds displayed no evident difference before and after 3-month storage. Also, no obvious change in the mechanical properties was identified before and after 3-month storage (n=5). (Ultimate stress: 1.31 ± 0.11 vs. 1.23 ± 0.24 MPa, $p > 0.05$; elastic modulus: 1.68 ± 0.07 vs. 1.74 ± 0.14 MPa, $p > 0.05$; elongation at break: $140.1 \pm 5.9\%$ vs. $132.3 \pm 9.6\%$, $p > 0.05$)

Figure 1. Optical images of the morphology and organization of extracellular matrix-derived scaffolds with aligned microchannels before and after 3-month storage. Transverse sections were stained with H&E staining.

Comment #9: Page 10, paragraph 2 At 1 day culture period is insufficient to evaluate cell migration, alignment and distribution. A more extended experiments needs to be performed, with different time-points and conditions. This will need to be included somehow in the main figures (maybe as part of Fig1) to give the reader a clear immediate view of the results. The evaluation of nuclear shape can be insufficient to indicate the circularity of the cells, at variance with i.e. the measurement of the cell shape and size through a cytoplasmic staining. More details on the technique utilized should be provided in the methods section.

Response: Thanks for the suggestion. We have extended the culture up to 7 days, and evaluated the cellular activities including morphology, migration, and spatial distribution for each time point (1, 3 and 7 days). In order to better demonstrate the guidance effect of parallel microchannels in ECM-derived scaffolds, we also evaluated the expression of some major genes in L6 cells, RSC96 cells and A10 cells after culture on the corresponding scaffolds for 7 days. More details on the methodology were provided in the revised manuscript.

Comment #10: Fig 2G, It's not clear if the force measurements have been performed before or after the EtOH sterilization, this needs to be specified as the process is known to alter the mechanical properties of the scaffolds.

Response: Sorry for not making this clear. The mechanical testing was performed after the ethanol sterilization, which was specified in Method section of the revised manuscript.

Comment #11: Page 11 The authors needs to specify the reason for choosing the Bovine Pericardium as a control scaffold. The article aims at repairing skeletal muscle, vascular and nerve defects and with the availability of decellularized scaffolds generated from every of these tissues, these could have been a better matched control. The total absence of capillaries within the pericardium after implantation, confirms that the scaffold is not an ideal control to assess cell migration. Similarly, in my opinion this bias the observations reported in Figure 3.

Response: We really appreciate the suggestion. As mentioned early, we now included the more relevant controls in our study to address the innovation of the ECM-derived scaffolds. As part of new experiments, *in vivo* cellularization, vascularization, macrophage response and regeneration of muscle, nerve and vascular tissue were re-evaluated and presented in the revised manuscript (see revised Figure 3-7).

Comment #12: Fig 3: Here is a bit confusing il4 and il10 have a signature which is more anti-inflammatory or at least immune-modulating. I was expecting higher values of these two for their scaffold. In particular il4, which is definitely an m2 associated cytokine. "...IL-4 (i), IL-10 (j)..."

Response: Sorry for the confusion. We totally agree that IL-4 and IL-10 would be cytokines associated with M2 anti-inflammatory phenotype. In this study, the host immune response to ECM-C scaffolds would be more favorable for M2 activation, which was confirmed by immunostaining for elevated CD206 (Fig. 4h). Indeed, materials-induced immunomodulation is essential for new tissue regeneration, which was specifically studied in our previous effort [4]. Since the immunomodulation by ECM-C was not the focus of the current study, we did not perform additional analyses of cytokines secreted by macrophage activation. Our ongoing research specifically looks into this, which will be presented in the future report.

[4] Zhu, M. et al. Biodegradable and elastomeric vascular grafts enable vascular remodeling. *Biomaterials* 183, 306-318 (2018).

Comment #13: Fig 4 The figure lacks the most basic controls. It shows the comparison of an untreated defects, with treated ones, in which any treatment would have in any case produced a better outcome than leaving an empty gap. The deposition of large amount of ECM components highlighted by the authors in figure 4F may indicate a fibrotic response. Please explain.

Response: Similar to the issue described above, appropriate controls for respective muscle, nerve and vascular repair were designed and used for the new experiments. The obtained results were included in the revised manuscript (now Fig. 5, Fig. 6 and Fig.7). With the new set of controls, the advantages of ECM-C in oriented tissue regeneration become obvious. A large amount of ECM deposited in ECM-C scaffolds as shown in Fig. 5d (Fig. 4f in the original submission) was further evaluated by Masson trichrome staining for collagen. Interestingly, much lower collagen was detected in the ECM-C treated muscle defect (Fig. 5d lower panel, Fig. 5e and Fig. 5f) compared to controls, indicating the non-fibrotic nature of the newly deposited ECM.

Comment #14: The volume recovery reported in 4G is compared to an unrepaired damage, therefore presenting a misleading result to the reader. The quantification reported in 4H indicates the % of fibers / area, this panel should also include the quantification of the number of fibers present in the area. The number of neuromuscular junctions / area presented in 4I and S5a require quantification as the author speculate on the meaning of the numbers detected in the control group. Overall the lack of appropriate controls makes these results severely incomplete. I suggest the authors repeat the experiments using a ECM control (i.e. pericardium or even better SIS), and quantify the results in comparison to their scaffold.

Response: Sorry for the confusion. In the revised manuscript, we have reorganized the presentation with additional experiment results and appropriate controls (see the new Fig. 5). Furthermore, we also included the quantification of muscle fibers presented in the area and the number of neuromuscular junctions in both ECM-C and control groups (FigS6).

Comment #15: Fig S8 The figure seem to be flipped, with native on the top (a-e) and neo-artery in (f-j). If this is not the case, the native vessel presented in I and J seems otherwise to be completely “de-endothelialized”. I feel that there is something wrong here. Figure 5 The red vWF staining presented in G is not visible, please present split-channel images as space is available in the figure.

Response: Sorry for using the unclear figures. We have changed the blurred images (Fig S8I, J and Fig 5G) with clear ones in revised manuscript (Fig.7i and Fig. S11).

Comment #16: Figure 6 Overall I think that the study is a bit confusing, the three experiments are performed with three different control setups, making a bit difficult to have an overall comparison, I would prefer to see the same controls in every experiment, it could be that they have the right controls, but they showed only the ones who makes their scaffold more appealing. It's not clear to me how the nerve regeneration was assessed. The staining presented in 6F is not clear, looks more

background and the recognition of the axons is hard to the sight. A higher magnification and an un-injured nerve staining will be required here.

Response: Thanks for the suggestion. In the revised manuscript, we have redesigned the control scaffolds and make them to be consistent. Meanwhile, additional experiments have been performed to evaluate the repairing capability of ECM-C in peripheral nerve regeneration. The unclear images were also replaced with clear ones at a higher magnification (see the new Fig. 6 of the revision).

Comment #17: It is difficult to understand how the number of regenerated axons visible in red in 6L could not possible sustain the level of regeneration presented in 6O.

Response: Sorry for the confusion, which might come from the unclear arrows in the images. In the revised Fig. 6, we corrected this. We found that the number of axons in control group was significantly lower than that of ECM-C scaffolds and autografts. Obvious atrophy and fibrosis of gastrocnemius muscle was detected in the control group while no significant difference in gastrocnemius muscle morphology was observed between ECM-C scaffolds and autograft.

Discussion

Comment #18: Page 30 The authors define the peritoneal cavity “a bioreactor”. It would use more broad terms such as “bioreactor-like site” or “scaffold-remodeling site”.

Response: Following the suggestion, we corrected this in the revised manuscript.

Comment #19: This section should be toned down or modified, the current results of decellularized materials are better than what is described here: “However, bovine pericardium, a popular material choice for use as a filler or to provide mechanical support for tissue defects, exhibited limited infiltration of host cells due to its dense pore structure. Indeed, a large number of reports have demonstrated that acellular matrices, such as small intestinal submucosa (SIS), urinary bladder, arteries, heart valves, the fascia lata, the dermis and tendon failed to support host cell infiltration, and led to inadequate tissue remodelling and functional integration^{14, 38-43}.”

Response: Based on the suggestion, we revised the section accordingly with more accurate description and insightful discussion of our research findings.

Comment #20: Page 33 The authors speculate on the use of their scaffolds in large animals and clinical studies. It needs to be mentioned in the discussion the problem of gas and nutrient diffusion in large volume of tissue / scaffolds. The last sentence of the page on reprogramming and gene editing is not clear, looks out of context and lacks of references.

Response: Thanks for pointing this out. Indeed, sufficient gas and nutrient exchange are necessary in regenerating large volume tissues. In recognition, we included the discussion on this issue in the revised manuscript. Furthermore, we also revised the description about gene editing/cell reprogramming with references.

Comment #21: This sentence is true, but the interleukins profile seems in contrast with this, please explain: “Compared with the bovine pericardium, ECM scaffolds with aligned microchannels favoured M2 macrophage phenotypic switching.”

Response: Thanks for the comments on this. The host is more tolerant to materials with good compatibility, and the balance between anti-inflammatory and pro-inflammatory factors is normally consistent with such responses, in our previous study on vascular regeneration, we had observed the close immunomodulation by materials [4]. With the redesign of controls in the revised manuscript, the manuscript has been reorganized with better streamline of the experimental results.

[4] Zhu, M. et al. Biodegradable and elastomeric vascular grafts enable vascular remodeling. *Biomaterials* 183, 306-318 (2018).

Comment #22: 22 and 34 is the same article

Response: We removed the repeated references in the revised manuscript.

Comment #23: Overall, the paper is interesting but confusing at time. The three experiments are performed with three different control setups, making a bit difficult to have an overall comparison; I am not sure bovine pericardium is the most appropriate control.

I think that they don't properly comment the potential clinical translation of this approach, which, to me, seems at least a bit complicate.

Response: Sorry for using the inappropriate controls in our initial submission. During the revision of current manuscript, we redesigned the controls and performed new *in vitro* and *in vivo* experiments. With the new results, we made a significant revision to the original submission to better elaborate the innovation, comprehensive analyses and in-depth discussion of the data. We also revised the discussion with potential implication of the reported approach for clinical translation.

Reviewer #2 (Remarks to the Author):

Comment #1: The authors describe the development and use of a novel scaffold derived from ECM but engineered to contain a “porous” channel network that allowed for greater cell in growth and organized alignment when used as a vascular, muscle and nerve template. Overall the experimental design is thorough, and it is commendable that the authors have chosen to demonstrate function in three distinct animal models to demonstrate functionality. Having said that, there are several points that the authors should address and clarify.

Response: We really appreciated the positive comments from the reviewer.

Comment #2: The authors talk in the introduction about porosity and the lack of pores in decellularized ECM materials and the need to increase porosity to support better remodeling. This is a very limited view and is certainly an engineer’s perspective and not a biologist’s perspective. When in place in the body as part of normal tissue the ECM has no pores, the pores that are observed by microscopy as generally an artifact of the scaffold being hydrated, normal tissues are not full of holes, those holes are filled with hydrated proteoglycans and other proteins that cells have to push through, there are no channels of pores in normal tissues and ECM. I would argue that in this study the authors are not developing a scaffold with increased porosity but have simply engineered an ECM scaffold to have aligned channels to promote cell migration, that is to say the ECM that forms around the template scaffold has a “porosity” that is independent of anything that authors are able to engineer. The leaching of the template scaffold simply creates channels in the matrix that act as conduits for cells the “porosity” of the actual ECM, if you could call it that, is unaffected. I would suggest that the authors consider revising their use of the term porosity and consider a better description for their hybrid scaffold as generating conduit channels rather than pores.

Response: Thanks for constructive comments. The ‘pore’ we mentioned in the manuscript is to defined the micron-size and even millimeter-size microchannels of the decellularized ECM scaffold with controllable distribution and alignment which can be engineered, but not the sub-micron size pores within the ECM fibers. Also, these microchannels and the porosity were defined and analyzed by microCT. To make this clear, we did use microchannels to replace the pores in the revised manuscript.

Comment #3: In the section on the immunomodulatory effects of the scaffolds I do not fully understand the choice of scaffolds and how the macrophage polarization results compare. The authors state that they used autologous, allogeneic and heterologous ECM scaffolds. It is not stated what how the autologous ECM scaffold was prepared, given that the production of the scaffold was 4-weeks not including the 7-8 day decellularization process. Can the authors be more definitive that this was decellularized ECM derived from the same animal? The allogeneic scaffolds and the heterogeneic scaffolds were also compared to bovine pericardium as a control. Given

that this is another example of a heterogeneous scaffold I feel the authors need to give more details as to why this particular scaffold acted as control. Was this based on ability to compare to prior data i.e. to confirm an appropriate host response was occurring or was this a true negative control that the authors knew would generate an unfavorable host response to which the others could be compared? There is a wealth of data already published demonstrating that xenogeneic and allogeneic ECM prepared from decellularized tissues does not elicit a foreign body reaction or any kind of immunological rejection response, so it seems excessive to perform these experiments on so many different ECM sources without discussing the relevance of the results in more detail.

Response: Sorry for the confusion partly from the inappropriate controls. Following the suggestion, we have unified the control group throughout the revised manuscript, removed irrelevant data and supplemented with a large portion of new data to improve the manuscript (see the revised manuscript).

Comment #4: One would argue that soaking in 75% ethanol for 12 hours is not a sterilization technique, but a disinfection technique, the authors do not demonstrate that the scaffolds meet sterility standards following this step.

Response: Thanks for the comments. We have corrected the inappropriate description in the method section of the revised manuscript. The scaffolds were sterilized by incubating in 75% ethanol for one hour, which was also used in other published reports [5, 6].

[5]. Xu, C.C., Chan, R.W., Weinberger, D.G., Efun, G. & Pawlowski, K.S. A bovine acellular scaffold for vocal fold reconstruction in a rat model. *Journal of Biomedical Materials Research Part A*: 92, 18-32 (2010).

[6] Keane, T.J., Swinehart, I.T. & Badylak, S.F. Methods of tissue decellularization used for preparation of biologic scaffolds and in vivo relevance. *Methods* 84, 25-34 (2015).

Comment #5: As previously state the use of ECM scaffolds and decellularized tissues for regenerative medicine is well established and there are a plethora of examples of their success, regardless of their lack of “pores” or microchannels. I feel it would be helpful in the authors could make a more definitive statement about why there is a need for engineered decellularized scaffolds. Can the authors provide examples where ECM scaffolds have failed as a result of lack of cell alignment or cellular infiltration? Or provide more specific descriptions of injuries that required an engineered scaffold or where microchannels would be necessary. I can see how microchannels would be beneficial for nerve conduits, but additional examples would be helpful for the reader.

Response: Thanks for the suggestion. As a matter of fact, the role of microchannel as a new porous structure was systematically described in a recent review [7] as follows: “Microchannels are perfusable architectural features engineered into biomaterials to promote mass transport of solutes to cells, effective cell seeding and compartmentalisation for tissue engineering applications, and survival, integration,

and vascularisation of engineered tissue analogues *in vivo*". Here, we demonstrated the guiding and promotive effect of the parallel-aligned microchannels on oriented tissue regeneration including muscle, nerve and artery. We have clarified the importance and the novelty of the decellularized ECM-derived scaffold with designed structure and the prior art of the decellularized scaffolds in both introduction and discussion sections. Relevant references were also included in the revised manuscript. [7] KS, L., M, B., S, M., TBF, W. & J, R.-K. Microchannels in Development, Survival, and Vascularisation of Tissue Analogues for Regenerative Medicine. Trends in biotechnology, (2019). DOI:10.1016/j.tibtech.2019.04.004.

Comment #6. The authors report that "Previous studies demonstrated that polymer-based matrices could be used as pore-forming templates to generate porous ECM scaffolds in vitro^{17, 30}, however, these scaffolds possessing low mechanical strength restrict their in vivo implantation." The authors, however, do not report the mechanical properties of these competitor scaffolds, nor do they report the mechanical strength that would be required for any of the native tissues into which they implanted their scaffolds. Without this information, it is very difficult to assess whether the mechanical properties of these new scaffolds are sufficient. Further, the authors (in Figure 2) only show the mechanical properties of one of their prepared scaffolds, but they report having generated three different types of scaffolds, each of which would ostensibly have unique mechanical properties. The authors should perform mechanical testing on all of their materials or explicitly state why they chose not to do so.

Response: As your comments, we describe improperly about the mechanical properties of reported scaffolds. We have modified the corresponding description and made the efforts to revise the manuscript completely. Also, we tested the mechanical properties of ECM-C scaffolds again with the new control scaffolds, as shown in Fig S4 of the revised manuscript. We think that the mechanical test of one type scaffolds (membrane) can represent the mechanical characteristic of the ECM-C group and the control group, so we did not tested the mechanics of the other two kinds of tubular scaffolds.

Comment #7: Similarly, the choice of control in their experiments may not be appropriate. The authors used bovine pericardium because it is a commonly used ECM product. This material, however, is not derived from the same species or the same anatomic location as the ECM from which the authors are generating their scaffolds. It would be ideal to utilize ECM derived from a subcutaneous pocket into which a sham material was placed, or no material was placed. In this way, comparisons between porosity, mechanical properties and cellular behavior can be made more readily and significant changes in cellular behavior can be directly attributed to the use of a sacrificial PCL scaffold, rather than differences in tissue type. Furthermore, a native sample should be included in Figure 2f.

Response: Thanks for the constructive comments. Based on the suggestion, we chose the appropriate controls in the revised manuscript to address the advantages of

ECM-C scaffolds in regenerating the oriented tissues. For generating the control scaffolds, silicon membranes or rods coated with a thin layer of PCL (thickness= about 12 μ m) were implanted subcutaneously for 4 weeks, and followed by the removal of polymer and cells to generate the decellularized control scaffolds. Thin PCL coating onto silicon membranes (for muscle repair) or rods (for nerve repair) was achieved by dipping into a solution of PCL in chloroform (0.1g/mL) and dried at room temperature. Additional *in vivo* and *in vitro* experiments were conducted to demonstrate the efficiency ECM-C scaffolds in the regeneration of oriented tissues.

Comment #8: On page 8 of the results section regarding cell alignment and cytocompatibility in vitro, the choice of control is again problematic. As a control, the authors use collagen scaffolds. This seems inappropriate because there are myriad other components in an ECM scaffold than just collagen, most of which will have bioactivity. It is more appropriate to compare any results against another ECM type such as dermis or urinary bladder ECM.

Response: Taking the advice, the control scaffolds were redesigned and evaluated for their regenerative capacity compared with the ECM-C group throughout the entire study.

Comment #9: It is unclear why the authors chose to generate their porous ECM scaffolds in different animal types (rat vs. rabbit), and at no point in the manuscript does this objective arise prior to pg. 8. The authors need to more thoroughly introduce this reasoning.

Response: Sorry for the confusion in the initial submission. With the new controls and experimental data, the manuscript was completely revised with better rationale for the experimental design (see the revised manuscript).

Comment #10: The functional test used to determine functional muscle tissue deposition, the jump test, is somewhat qualitative. Furthermore, 30 days is not a long enough period of time for mature muscle fiber deposition, with 60 days being a more appropriate timepoint. More rigorous tests would have included isometric torque analysis plus EMG. The authors should ideally perform these studies, as well, or at least acknowledge the limited scope of their chosen functional tests.

Response: Thanks for the suggestion. Although we observed an obvious difference in muscle tissue regeneration between the experimental and control groups in 30 days, we still plan to evaluate the long-term regeneration effect in rat model and large animal experiments in the future studies. Also, we acknowledged the limited scope of the chosen functional tests in the discussion section of revised manuscript.

Comment #11: Within the same section, pg. 13, the authors state that “decellularized muscle tissue scaffolds were also implanted subcutaneously, showing few cell infiltration and vascularization with sparse distribution of an amount of CD206 positive cells and some capillaries around the border of the scaffolds.” The reviewer

is left asking why this study was done, what “decellularized muscle tissue scaffolds” represents, and what controls were performed in tandem with this material? Greater detail is required when discussing this experiment and its results, as well as a more uniform system of reference for the materials used throughout this manuscript.

Response: Sorry for the confusion. In our original submission, the controls were not appropriately designed. In this regard, new control scaffolds were fabricated and included to evaluate the advantages of ECM-C, which was presented in the revised manuscript.

Comment #12: For the vascular results, the results section should include which arteries these scaffolds were implanted into, and what the patency of the ECM scaffolds is being compared to (are they compared to native controls?). Furthermore, the results showing that patency was nearly 100% is surprising and should be placed in line with previously reported results. These results are not addressed later in the discussion.

Response: Thanks for your comments. The abdominal aortic defect in rats were used to evaluate the vascular regeneration of ECM-C scaffold. We included this in results section of the revised manuscript. The patency of the ECM-C scaffolds (11/11) are as follows: the ratio of the number of unobstructed ECM-C vascular scaffolds to the number of all implanted ECM-C scaffolds. We supposed that the high patency of the implanted ECM-C scaffolds was mainly because of complete endothelialization at three months, which was consistent with our previous study [4]. We also added the corresponding explanation in discussion section of revised manuscript.

[4] Zhu, M. et al. Biodegradable and elastomeric vascular grafts enable vascular remodeling. *Biomaterials* 183, 306-318 (2018).

Comment #13. For the nerve resection, the size of the defect should be included in the results section.

Response: Thanks for the advice, we included the size of nerve defect (15mm) in the results section of the revised manuscript.

Comment #14. Any time a significant difference is noted, the p-value needs to be reported.

Response: Thanks for the suggestion, and we included the p-value in the revised figures with a significant difference.

Comment #15: In the methods section, more detail is needed regarding antibodies used for immunostaining. In particular, the composition of blocking antibodies, the duration of incubation with blocking buffer, and the dilutions of antibodies are all necessary.

Response: More details were provided to the method section of the revised manuscript .

Comment #16. “polylactide acid (PLA)” should be changed for “Polylactic acid (PLA)”

Response: Thanks for pointing this out and we made changes accordingly.

Comment #17. In the sentence “have been reported to induce acute, and in some cases,” change “is for “in”

Response: Sorry for the typo, and we seek editorial assistance for the revised manuscript.

Comment #18. In the phrase “Biocompatible and porous ECM scaffolds” correct the underlined word for “Biocompatible”

Response: This was corrected in the revision.

*Comment #19. The authors claim that “no significant difference was found between the heterologous ECM and pericardium scaffolds (Fig. 3f)”, however, the graph that compares those 2 groups has an * (p<0.05) indicating significant differences. This should be corrected.*

Response: The inconsistency was corrected in the revised manuscript.

Comment #20.: “CMAP” is mentioned for the first time in the caption of figure 4, but the acronym description occurs later in the paper: “The ratio of the compound muscle action potentials (CMAP)...”

Response: In the revised manuscript, we have spelled out the acronym.

Comment #21. In some cases, there are 2 spaces between words.

Response: We have corrected it in the revised manuscript.

Reviewer #3 (Remarks to the Author):

The authors designed a extracellular matrix-based bioscaffold with good biocompatibility and low immune response and examined the in vivo effect of the novel bioscaffold. Their outcomes powerfully showed that the novel bioscaffold could guide cell alignment, promote cellulization, encourage vascularization, regulate macrophage M1/M2 state switch. Functional outcomes demonstrated that the bioscaffold was able to promote the regeneration of skeletal muscle, arterial tissue, and nerves.

The study is integrated and well-designed. Statistical methods are suitable used as well. One minor comment is that the authors should study or at least discuss more about the underlying cellular and mechanisms of the promoting role of extracellular matrix-based bioscaffold in tissue engineering. I would suggest "accept with minor revision".

Response: We really appreciate the very positive comments. In this revised manuscript we added additional discussion on the underlying cellular and mechanisms for promoting tissue regeneration by ECM-derived scaffolds with parallel microchannels (see the revised discussion).

REVIEWERS' COMMENTS:

Reviewer #1 (Remarks to the Author):

Thank you for the revised manuscript which looks much improved. The authors have response to my comments and questions.

Reviewer #2 (Remarks to the Author):

ACCEPT

I have no further comments or revisions. The authors have addressed the limitations of the original draft and added the requested information.